

# Superstring amplitudes meet surfaceology

**Qu Cao[1,2⋆], Jin Dong[2,3†], Song He[2,4,5‡] and Fan Zhu[2,3,4∘]**

**1** Zhejiang Institute of Modern Physics, School of Physics,
Zhejiang University, Hangzhou, 310027, China
**2** Institute of Theoretical Physics, Chinese Academy of Sciences, Beijing 100190, China
**3** School of Physical Sciences, University of Chinese Academy of Sciences,
Beijing 100049, China
**4** School of Fundamental Physics and Mathematical Sciences,
Hangzhou Institute for Advanced Study, UCAS, Hangzhou 310024, China
**5** International Centre for Theoretical Physics Asia-Pacific, Beijing 100190, China

⋆ qucao@zju.edu.cn , † dongjin@itp.ac.cn , ‡ songhe@itp.ac.cn , ∘ zhufan22@mails.ucas.ac.cn

## Abstract

We reformulate tree-level amplitudes in open superstring theory (type-I) in terms of stringy $\text{Tr}(\phi^3)$ amplitudes with various kinematical shifts in the "curve-integral" formulation: while the bosonic-string amplitude with $n$ pairs of "scaffolding" scalars comes from a particularly simple shift of the $\text{Tr}(\phi^3)$ one (corresponding to $n$ length-2 cycles), the analogous superstring amplitude requires "correction" terms given by bosonic-string amplitudes with longer, even-length "cycles", which are also $\text{Tr}(\phi^3)$ ones at shifted kinematics dictated by the cycles; in total it is expressed as a sum of $(2n-3)!!$ shifted amplitudes originated from the expansion of a reduced Pfaffian. Upon taking $n$ scaffolding residues, this leads to a new formula of the $n$-gluon superstring amplitude, which is manifestly symmetric in $n-1$ legs, as a gauge-invariant combination of mixed bosonic string amplitudes with gluons and scalars, which come from length-2 cycles and longer ones respectively (the total sum is associated with the expansion a $n \times n$ symmetrical determinant); the corresponding prefactors are nested commutators of $2n$-gon kinematical variables, which nicely become traces of field-strengths for those legs corresponding to scalars in the mixed amplitudes. These interesting linear combinations of bosonic string amplitudes must guarantee the cancellation of tachyon poles and $F^3$ vertices *etc.*, and they give new relations between the superstring amplitude and its bosonic-string building blocks to all orders in the $\alpha'$ expansion (the first order gives a new formula for gluon amplitudes with a single $F^3$ insertion in terms of Yang-Mills-scalar amplitudes). We provide both the worldsheet and "curve-integral" derivations, and discuss applications to heterotic and type II cases.

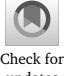

# 1 Introduction

Recently, a new formulation has been proposed for scattering amplitudes in the simplest colored-scalar theory, the Tr($\phi^3$) theory, to all loops and to all orders in the 't Hooft coupling, based on "curve integrals" on surfaces [1–3] (see [4–8] for earlier works). Subsequently in [9] (see also [10, 11] it has been shown that by simple kinematical shifts, these curve integrals also compute all-loop scattering amplitudes in non-linear sigma model (NLSM) and scalar-scaffolded Yang-Mills theory (YM), which thus represent a novel unity of the simplest colored scalars, pions and gluons, and make some of their hidden properties manifest, such as zeros and splittings near zeros [11–13]. At tree level, the curve-integral formula for Tr($\phi^3$) amplitudes is equivalent to certain disk integrals for open strings (known from early days of dual resonance model [14] and also dubbed as $Z$ integrals in [15–18])), and the result of [9] including zeros and splits automatically apply and generalize for *stringy* amplitudes of colored scalars, pions and gluons. As we will review shortly, the curve-integral for stringy Tr($\phi^3$) amplitude with $2n$ colored scalars has the integrand of the form $\prod \frac{dy}{y}$, which corresponds to the famous "Parke-Taylor" disk integrand $1/(z_{1,2} \cdots z_{2n-1,2n} z_{2n,1})$, in addition to the universal Koba-Nielsen factor [19], $\prod u^X$; after performing the shift that gives the scalar-scaffolded $n$-gluon tree amplitude, it gives nothing but the bosonic string amplitudes for $n$ pairs of scalars *e.g.* $(1,2),(3,4),\ldots,(2n-1,2n)$, (which is the coefficient of $\epsilon_1 \cdot \epsilon_2 \epsilon_3 \cdot \epsilon_4 \cdots \epsilon_{2n-1} \cdot \epsilon_{2n}$ of $2n$-gluon bosonic string amplitudes [20]); the "curve-integrand" from the shift becomes $\prod \frac{dy}{y^2}$, which nicely corresponds to the disk integrand $1/(z_{1,2}^2,\ldots,z_{2n-1,2n}^2)$. Therefore, this shift, which turns the amplitude of stringy Tr($\phi^3$) into that of $n$ pairs of scalars in bosonic string theory, amounts to the shift that turns a single length-$2n$ "cycle", $(12\cdots 2n)$ into $n$ length-2 cycles,$(12),(34),\ldots,(2n-12n)$; as explained in [20], important properties of gluon amplitudes in this $2n$-scalar language, such as the gauge-invariance (and multilinearity in polarizations), as well as factorizations on gluon poles, can be shown directly at the level of curve integrals (which follow from combinatorial properties of $\int \prod \frac{dy}{y^2} \prod u^X$; after taking the scaf-

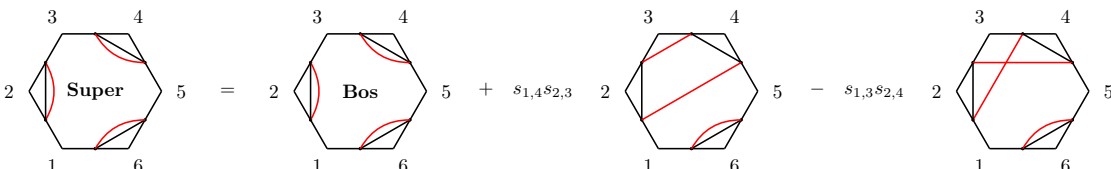

Figure 1: Example of 6-gon superstring amplitude. Red line denotes the perfect matching $\frac{1}{z_{i,j}}$. Black line denotes overall factor $\frac{1}{z_{2i-1,2i}}$. Here we have omitted $s_{1,2}-1$ and $s_{3,4}-1$.

folding residues with $(p_1+p_2)^2 = \cdots (p_{2n-1}+p_{2n})^2 = 0$ (where each pair of scalars become an on-shell gluon), the resulting "curve-integrand" turns into exactly the $n$-gluon bosonic-string correlator given by the usual OPE of vertex operators.

Of course the truly remarkable point of [20] is that there is strong evidence suggesting that the same procedure at higher-genus surfaces directly leads to certain "stringy" completion of all-loop gluon amplitudes in scalar-scaffolded language, where gauge invariance and factorization/cut properties for such "canonical" integrand of YM loop amplitudes based on "surface kinematics" can be shown in a similar way [3]. However, already at tree level, an obvious question concerns superstring theory in this curve-integral formulation: while the worldsheet coordinates, $z$ variables, are naturally replaced by $y/u$ variables associated with curves on the $n$-punctured disk, what about the Grassmann variables associated with worldsheet fermions, and what do combinations like $z_i - z_j + \theta_i \theta_j$ correspond to in this language? Since currently we do not know any intrinsic formulation of the worldsheet supersymmetry directly in terms of curve integrals, in this note we aim at a warm-up exercise of rewriting the tree-level superstring amplitudes in the RNS formalism [21–23] in this new formulation, and we do so first for $n$ pairs of scalars, which, upon taking scaffolding residues, produce $n$ gluons in superstring theory. Interestingly, even this simple exercise will teach us several lessons. First of all, unlike the bosonic string amplitude for $n$ pairs of scalars, which is given by *one term*, $\prod \frac{dy}{y^2}$, obtained from the simplest shift, the superstring amplitude turns out to be a sum of $(2n-3)!!$ terms: the simplest term again follows from the simplest shift which gives $\prod \frac{dy}{y^2}$, and any of the "correction" terms has its own kinematic shifts from the stringy $\text{Tr}(\phi^3)$ amplitude; very nicely, we will see that all these terms come from the expansion of a reduced Pfaffian of $2n \times 2n$ antisymmetric matrix on the worldsheet, where each term has its own cycle structures with even-length cycles. For example, for $n = 3$, we have two "correction" terms $\frac{s_{14}s_{23}}{z_{1,4}z_{2,3}z_{1,2}z_{3,4}z_{5,6}^2}$ (with cycles $(1234)(56)$) and $\frac{s_{13}s_{24}}{z_{1,3}z_{2,4}z_{1,2}z_{3,4}z_{5,6}^2}$ (with cycles $(1243)(56)$), both of which can be obtained from certain kinematic shifts of the stringy $\text{Tr}(\phi^3)$ amplitude just as the original bosonic string term proportional to $\frac{1}{z_{1,2}^2 z_{3,4}^2 z_{5,6}^2}$ (with cycles $(12)(34)(56)$).

Note that there is a reflection of the underlying worldsheet supersymmetry in that we choose to delete two columns/rows corresponding to a pair, *e.g.* $(2n-1, 2n)$, which correspond to the two scalars in $-1$ ghost picture. The upshot is that superstring amplitude is a sum of stringy $\text{Tr}(\phi^3)$ amplitudes with different kinematical shifts dictated by such cycle structures, each dressed with some prefactors of Mandelstam variables. We will see that these kinematic shifts are given by *combinatorial intersection* of any curve $(ab)$ with the corresponding cycle structure, *e.g.* for the original bosonic-string cycles $(12)(34)\ldots$, the intersection is 1 for $a, b$ odd, $-1$ for $a, b$ even and 0 otherwise; while for $(1234)(56)$ we have non-zero intersections 1 for $(a, b) = (2, 6), (3, 6), (4, 6)$ and $-1$ for $(a, b) = (1, 5)$ (see Fig. 1). Important properties such as gauge invariance, factorizations and even zeros and splits can be derived using the expansion just as in [20].

Moreover, we will see that upon taking scaffolding residues, this actually implies a new formula for $n$-gluon superstring amplitudes, as linear combinations of mixed bosonic string amplitudes with gluons and scalars, *e.g.* while for any 2-cycle such as $(12)$ with singularity $\frac{dy_{13}}{y_{13}^2}$, the residue at $y_{13} = 0$ produces a gluon, for any longer cycle such as $(1234)$ the singularity goes like $\frac{dy_{13}dy_{35}}{y_{13}y_{35}}$ near $y_{13} = y_{35} = 0$ which gives two scalars instead. In any such term, the gluon polarization for any leg which is a scalar in the mixed amplitude turns out to be given by the prefactors (of $2n$-gon kinematics): we will see that upon scaffolding these prefactors combine into certain *nested commutators* of monomials of $2n$-gon Mandelstam variables, which are nicely gauge invariant by themselves. In fact, they turn out to be (products of) traces of field-strengths, thus we have a manifestly gauge-invariant formula for $n$-gluon superstring amplitude in terms of mixed, bosonic string amplitudes (with strictly less number of gluons). For example, the two prefactors of $n = 3$ combine as $s_{1,4}s_{2,3} - s_{1,3}s_{2,4}$, which in terms of polarizations of the two gluons become $\text{tr}(f_1 f_2)$, thus the 3-gluon superstring amplitude is given by the bosonic string one, plus $\text{tr}(f_1 f_2)A_3^{\text{mixed}}((12)_s|3_g)$ where the mixed amplitude has scalar legs $1, 2$ (from scaffolding $(1, 2), (3, 4)$) and gluon leg 3 (from scaffolding the fixed pair $(5, 6)$).

This new formula represents a novel relation between superstring amplitude and all the mixed bosonic-string ones, independent of whether it is derived from curve-integral or the worldsheet. One novelty of our result is that the pair of scalars in $-1$ picture leads to only one special gluon, *e.g.* leg $n$, thus it is manifestly symmetric in all other $n-1$ legs (which is not the case of usual representations *e.g.* with two gluons in $-1$ picture). Note that the combination of these bosonic-string amplitudes is very special, which must guarantee the cancellation of tachyon poles, and also the $F^3$ vertices which cannot appear in the superstring amplitudes. For the $n = 3$ example, we see that the correction amounts to $-\text{tr}(f_1 f_2 f_3)$ which cancels the $F^3$ vertices of the bosonic one and gives the correct superstring answer, or the 3-gluon YM amplitude. We will show that the combinatorics of the sum is exactly given by the expansion of an $n \times n$ symmetric determinants, where in each term the product of off-diagonal entries form cycles for the scalars and that of the diagonal entries represent remaining gluons (in bosonic string amplitudes); there are $2, 5, 17, 73, 388, \ldots$ such terms for $n = 3, 4, 5, 6, 7, \ldots$, which are relatively economic.

We will also see that the formula implies new relations between the $n$-gluon superstring amplitude and its bosonic string building blocks, to all orders in the $\alpha'$ expansion. Note that at leading order, both superstring and bosonic string amplitudes with $n$ gluons are given by the Yang-Mills amplitude, which is consistent with the fact that the $\alpha'$ expansion of all mixed amplitudes starts at higher orders; at next order $\mathcal{O}(\alpha')$, the $n$-gluon bosonic string amplitude reduces to the YM amplitude with a single $F^3$ insertion, while the superstring amplitude vanishes since it starts at $\mathcal{O}(\alpha'^2)$. This implies a new formula of the YM amplitude with a single $F^3$, as a sum of mixed amplitudes dressed with traces of field-strengths at $\mathcal{O}(\alpha')$; it turns out that only YM-scalar amplitudes with a single trace of scalars contribute at this order. Of course, we also find new relations at higher orders $\mathcal{O}(\alpha'^k)$ for $k > 1$, which in general relate different contributions from (mixed) bosonic string amplitudes and that from the superstring amplitude at this order. Last but not least, our new representation for the superstring correlator can also be used for heterotic string and type-II case, which imply similar relations for closed-string amplitudes. Among other things, they also imply various new relations for gravitational amplitudes via double copy. All these results can be checked *e.g.* using Cachazo-He-Yuan (CHY) formulas [24, 25] for these field-theory amplitudes.

The paper is organized as follows. In sec. 2 we present our formula for the superstring amplitude with $n$ pairs of scalars as a sum of shifted stringy $\text{Tr}(\phi^3)$ amplitudes, and that for the $n$-gluon superstring amplitude as a linear combination of mixed bosonic-string amplitudes. In sec. 3 we study various properties of superstring amplitudes using the new formulas, including

new relations for the $\alpha'$ expansion including a new formula for the $F^3$ amplitude, new formulas for closed superstring amplitudes: heterotic and type II, under field-theory limit, also obtain the new relations for the $R^2$ and $R^3$ amplitudes. We provide detailed derivations based on the worldsheet formulation in sec. A and the "curve-integral" formulation in sec. B.1.

## 2 A unity of amplitudes for stringy $\mathrm{Tr}(\phi^3)$, bosonic string and superstring

In this section, we first present a new formula for the superstring amplitude with $n$ pairs of scalars, in a form similar to the scalar-scaffolded bosonic string amplitudes. We then take scaffolding residues and obtain a new formula for the $n$-gluon superstring amplitude, expressing the result as a gauge-invariant combination of mixed bosonic string amplitudes involving gluons and scalars, arising from length-2 cycles and longer cycles, respectively.

### 2.1 $2n$-gon superstring as a sum of shifted stringy $\mathrm{Tr}(\phi^3)$

Recall that the bosonic string amplitudes with $n$ pairs of "scaffolding" scalars can be obtained as the stringy $\mathrm{Tr}(\phi^3)$ amplitudes with a simple kinematical shift on the *planar variables* $X_{a,b} := \alpha'(p_a + p_{a+1} + \ldots + p_{b-1})^2$ of the $2n$-gon:[1]

$$\mathcal{A}^{\text{bos.}}_{(1,2),(3,4),\ldots,(2n-1,2n)} = \mathcal{A}^{\phi^3}_{2n}\left(X_{a,b} \to X_{a,b} + \delta^{e,o}_{a,b}\right), \tag{1}$$

where the famous "even/odd" shifts are defined as $\delta^{e,o}_{a,b} = 1$ for $a, b$ odd, $-1$ for $a, b$ even and $0$ otherwise, and the unshifted stringy $\mathrm{Tr}(\phi^3)$ amplitude is defined as:

$$\mathcal{A}^{\phi^3}_{2n} = \int_{D(12\cdots n)} \frac{dz_1 \cdots dz_{2n}}{\text{vol SL}(2,\mathbb{R})}[1,2,\ldots,2n]\prod_{i<j}z_{i,j}^{s_{i,j}} = \int_{\mathbb{R}^{2n-3}>0}\prod_J \frac{dy_J}{y_J}\prod_{(a,b)}u_{a,b}^{X_{a,b}}, \tag{2}$$

where we define $z_{i,j} = z_i - z_j$. Here the Mandelstam variables are $s_{i,j} := \alpha'(p_i + p_j)^2 = 2\alpha' p_i \cdot p_j$ with the standard conventions. The integration domain is the positive part of the real moduli space, $\mathcal{M}^+_{0,n}$, or $z_1 < z_2 < \cdots < z_{2n}$ (with 3 of them fixed by the SL$(2,\mathbb{R})$ redundancy). In the second equality we use the positive parametrization and rewrite the amplitude as a curve integral, where the product over $J$ runs over a given triangulation of the surface, *i.e.* a disk with $2n$ marked points on the boundary (see also a brief review in Appendix B.1). $[I]$ denotes a "Parke-Taylor" factor for any length-$k$ cycle $I = (i_1, \ldots, i_k)$,

$$[I] := \frac{1}{(z_{i_1} - z_{i_2}) \cdots (z_{i_k} - z_{i_1})}. \tag{3}$$

The second product in (2) runs over all possible chords on the surface, labeled by two marked points $a$ and $b$ satisfying $a < b$ and $b - a > 1$. The variable $u_{a,b}$, associated with the chord $(a,b)$, is a rational function of the $y_I$'s, completely determined by the geometry/combinatorics of the surface. These $u$ variables satisfy the $u$-equations

$$u_{a,b} + \prod_{(c,d) \text{ cross } (a,b)} u_{c,d} = 1, \tag{4}$$

which encode the structure known as "binary geometry". Remarkably, the solutions to these equations are given by:

$$u_{a,b} = \frac{z_{a-1,b}z_{a,b-1}}{z_{a-1,b-1}z_{a,b}}. \tag{5}$$

---

[1] In this paper, we ignore the overall dependence on the inverse string tension $\alpha'$.

These solutions provide a direct relation between the $u$-variables and the worldsheet coordinates $z$, a connection that will play an important role shortly.

The kinematical shift in (1) is specified by its correlator as the unshifted stringy $\mathrm{Tr}(\phi^3)$ multiplies a ratio of $z_{i,j}$, or equivalently, a ratio of $u$ variables:

$$
\begin{aligned}
\mathcal{A}^{\text{bos.}}_{(1,2),\dots,(2n-1,2n)} &= \int_{D(1\cdots 2n)} \frac{\mathrm{d}z_1 \cdots \mathrm{d}z_{2n}}{\mathrm{vol}\,\mathrm{SL}(2,\mathbb{R})} [1,2][3,4]\dots[2n-1,2n] \prod_{i<j} z_{i,j}^{s_{i,j}} \\
&= \int_{D(1\cdots 2n)} \frac{\mathrm{d}z_1 \cdots \mathrm{d}z_{2n}}{\mathrm{vol}\,\mathrm{SL}(2,\mathbb{R})} [1,2,\dots,2n] \prod_{i<j} z_{i,j}^{s_{i,j}} \frac{z_{2,3}z_{4,5}\cdots z_{2n-2,2n-1}}{z_{1,2}z_{3,4}\cdots z_{2n-1,2n}} \\
&= \int_{\mathbb{R}^{2n-3}>0} \prod_J \frac{d y_J}{y_J} \prod_{(a,b)} u_{a,b}^{X_{a,b}} \frac{\prod_{(e,e)} u_{e,e}}{\prod_{(o,o)} u_{o,o}},
\end{aligned}
\tag{6}
$$

where we take the product of all $u_{a,b}$ with $a, b$ both being even, over the product of $u_{a,b}$ both being odd. In the third line we have used $u_{a,b} = \frac{z_{a-1,b}z_{a,b-1}}{z_{a-1,b-1}z_{a,b}}$ to express the ratio of $z_{i,j}$ as a ratio in the $u$ variables. As we have shown in Appendix B.2, any general ratio of the form $\prod_{i,j} z_{i,j}^{d_{i,j}}/[1,2,\dots,2n]$ with a specified weight $-2$ for each $z_i$ (*i.e.* for integer exponents $d_{i,j}$ satisfying $\sum_{j\neq i} d_{i,j} = -2$), can be expressed as a ratio of monomials in the $u$ variables, which determines the corresponding kinematical shift:

$$
\frac{\prod_{i,j} z_{i,j}^{d_{i,j}}}{[1,2,\dots,2n]} = \prod_{(a,b)} u_{a,b}^{x_{a,b}}, \quad \text{with} \quad x_{a,b} = b - a - 1 + \sum_{a \le i < j < b} d_{i,j}.
\tag{7}
$$

Let us spell out some examples that would be useful later:

$$
\frac{z_{1,2}^{-1}z_{3,4}^{-1}z_{5,6}^{-2}z_{1,3}^{-1}z_{2,4}^{-1}}{[1,2,3,4,5,6]} = \frac{u_{2,4}u_{2,6}u_{3,6}u_{4,6}}{u_{1,5}},
\tag{8}
$$

where we have used $x_{2,4} = 1 + d_{2,3} = 1$, $x_{1,5} = 3 + d_{1,2} + d_{1,3} + + d_{1,4} + d_{2,3} + d_{2,4} + d_{3,4} = -1$ and so on. Similarly, we have

$$
\frac{z_{1,2}^{-1}z_{3,4}^{-1}z_{5,6}^{-2}z_{1,4}^{-1}z_{2,3}^{-1}}{[1,2,3,4,5,6]} = \frac{u_{2,6}u_{3,6}u_{4,6}}{u_{1,5}}.
\tag{9}
$$

Now let us consider the superstring amplitudes in the RNS language(in this paper, we only focus on the NS sector for external states). For $2n$-point scalar superstring amplitude, we can extract it from the $2n$-point gluon superstring amplitude by the coefficient of $\epsilon_1 \cdot \epsilon_2 \epsilon_3 \cdot \epsilon_4 \cdots \epsilon_{2n-1} \cdot \epsilon_{2n}$. We put the derivation in the Appendix A.2, and get the key building blocks of the $2n$-gon superstring amplitudes, the reduced Pfaffian of matrix $\tilde{\mathbf{A}}$. Here we define the $2n \times 2n$ antisymmetric matrix

$$
\tilde{\mathbf{A}}_{i,j} := \begin{cases} \frac{\tilde{s}_{i,j}}{z_i - z_j}, & i \neq j, \\ 0, & i = j, \end{cases}
\tag{10}
$$

where we also define

$$
\tilde{s}_{1,2} := s_{1,2} - 1, \qquad \tilde{s}_{3,4} := s_{3,4} - 1, \qquad \dots, \qquad \tilde{s}_{2n-1,2n} := s_{2n-1,2n} - 1,
\tag{11}
$$

and $\tilde{s}_{i,j} = s_{i,j}$ otherwise. As we will see, it makes sense to compute the reduced Pfaffian where we delete the two columns and rows $2n-1, 2n$ and define

$$
\mathrm{Pf}'\tilde{\mathbf{A}} := \frac{(-1)^{n-1}}{z_{2n-1} - z_{2n}} \mathrm{Pf}|\tilde{\mathbf{A}}|_{2n-1,2n},
\tag{12}
$$

where the overall sign $(-1)^{n-1}$ is chosen to compensate for the sign in $\prod(s-1)$, such that the pure bosonic part, *i.e.* $\frac{1}{z_{1,2}z_{3,4}\cdots z_{2n-1,2n}}$ has coefficient 1. The result is a sum over $(2n-3)!!$ perfect matchings as above, where for each $\alpha$ we define $S_\alpha$ to be the signed product of Mandelstam variables, $\text{sign}_\alpha \prod_{(i,j)\in\alpha} \tilde{s}_{i,j}$ (note the replacement $s_{2i-1,2i} \to s_{2i-1,2i}-1$). Each $\tilde{s}_{i,j}$ is accompanied by a factor of $z_i - z_j$ with $i < j$, and the sign arises from the Pfaffian expansion.

We show that the superstring amplitude with $n$ pairs of scalars is given by $\text{Pf}'\tilde{\mathbf{A}}$ multiplies by a factor $1/(z_{1,2}z_{3,4}\cdots z_{2n-1,2n})$ in the Appendix A.2. Therefore, on the support of (7), it reduces to a linear combination of stringy $\text{Tr}(\phi^3)$ amplitudes at different shifts, which correspond to amplitudes with longer cycles (interpolating between $\text{Tr}(\phi^3)$ and bosonic string with $n$ pairs of scalars):

$$\mathcal{A}^{\text{super.}}_{(1,2),\dots,(2n-1,2n)} = \sum_{\text{p.m. }\alpha}^{(2n-3)!!} S_\alpha \, \mathcal{A}^{\phi^3}_{2n}\left(X_{a,b} \to X_{a,b} + \delta^\alpha_{a,b}\right), \tag{13}$$

where both $S_\alpha$ and $\delta^\alpha$ can be obtained from expanding a Pfaffian abstractly. The sum is over all *perfect matchings*, denoted by $\alpha$, of $1,2,\dots,2n-2$ together with the fixed pair $(2n-1,2n)$ (which is chosen to be special as will be explained shortly), the prefactor $S_\alpha$ is given by a signed product of Mandelstam variables $s_{i,j}$, and $\delta^\alpha$ is the kinematical shift associated with $\alpha$. And $\delta^\alpha$ is the shift corresponding to the cycle factor (product of "Parke-Taylor" factors) defined as

$$C_\alpha := \prod_{i \text{ odd}} \frac{1}{(z_i - z_{i+1})} \prod_{(ij)\in\alpha} \frac{1}{(z_i - z_j)} = \text{sgn}'(\alpha) \prod_{I:=(i_1,\dots,i_k)} [I], \tag{14}$$

where $\text{sgn}'(\alpha)$ is a sign factor that compensates for the additional signs introduced by $\prod[I]$, since it contains $z_{i',j'}$ with $i' > j'$.[2] Note that for the special $\alpha_0 = (1,2),\dots,(2n-1,2n)$, $\delta^{\alpha_0}$ is given above and $S_{\alpha_0} = (s_{1,2}-1)(s_{3,4}-1)\cdots(s_{2n-3,2n-2}-1)$ gives rise to the original bosonic string term. Let us spell out details for $n = 3, 4$. For $n = 3$ there are 3 matchings (denoted as $\alpha_{0,1,2}$): in addition to the fixed pair $(5,6)$, the remaining pairs are $\{(1,2),(3,4)\}$, $\{(1,3),(2,4)\}$ and $\{(1,4),(2,3)\}$ respectively; the corresponding prefactors and cycle factors read

$$\begin{aligned} S_{\alpha_0} &= (s_{1,2}-1)(s_{3,4}-1), & S_{\alpha_1} &= -s_{1,3}s_{2,4}, & S_{\alpha_2} &= s_{1,4}s_{2,3}, \\ C_{\alpha_0} &= -[12][34][56], & C_{\alpha_1} &= -[1243][56], & C_{\alpha_2} &= [1234][56]. \end{aligned} \tag{15}$$

For $n = 4$, the $5!! = 15$ matchings are denoted as $\alpha_i$ for $i = 0, 1, \dots, 6, 7, \dots, 14$, where for $\alpha_1, \dots, \alpha_6$ the prefactors are $s_{1,3}s_{2,4}(s_{5,6}-1)$, $s_{1,4}s_{2,3}(s_{5,6}-1)$ and 4 more with $1,2$ or $3,4$ replaced by $5,6$, and for $\alpha_7, \dots, \alpha_{14}$ the 8 prefactors are given by products of three $s_{i,j}$ such as $s_{1,3}s_{2,5}s_{4,6}$; examples of the corresponding cycle factors are depicted in Fig. 2.

The corresponding shifts $\delta^\alpha$ can be easily read from $C_\alpha$: for each chord $(i,j)$, the shift $\delta^\alpha_{i,j}$ is given by its *combinatorial intersection* (7) with the cycles in $C_\alpha$. For example, for $C_\alpha = [12\cdots 2n-2][2n-1\,2n]$, the shift is $\delta^\alpha_{i,j} = 1$ for $i = 1, j = 2n-1$, $-1$ for $j = 2n$ ($\forall i$), and 0 otherwise. In this way, for $n = 3$, we obtain (see Eq. (8) and (9)):

$$\begin{aligned} \mathcal{A}^{\text{super.}}_6 &= (s_{1,2}-1)(s_{3,4}-1)\mathcal{A}^{\text{bos.}}_6 - s_{1,3}s_{2,4}\mathcal{A}^{\phi^3}(\delta_{2,4} = \delta_{2,6} = \delta_{3,6} = \delta_{4,6} = -\delta_{1,5} = 1) \\ &\quad + s_{1,4}s_{2,3}\mathcal{A}^{\phi^3}(\delta_{2,6} = \delta_{3,6} = \delta_{4,6} = -\delta_{1,5} = 1). \end{aligned} \tag{16}$$

For $n = 4$, in addition to the original bosonic string $\mathcal{A}^{\text{bos.}}_8$ with a factor of the form $(s-1)(s-1)(s-1)$, we have 6 terms with with a factor of the form $ss(s-1)$ and 8 terms

---

[2]Note that if we express the $2n$-scalar amplitude as a weighted sum of shifted $\text{Tr}(\phi^3)$, the factor $\text{sgn}'(\alpha)$ is absent by definition.

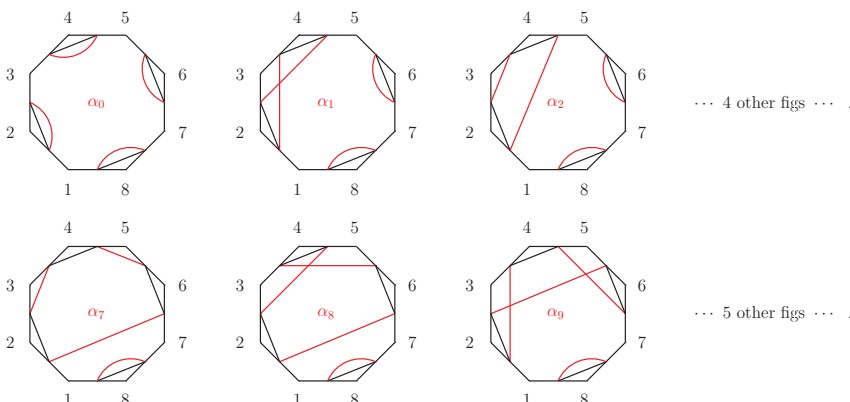

Figure 2: Examples for the perfect macthings of $n = 4$.

with a factor of the form $sss$:

$$
\begin{aligned}
\mathcal{A}_8^{\text{super.}} = {} & -(s_{1,2}-1)(s_{3,4}-1)(s_{5,6}-1)\mathcal{A}_8^{\text{bos.}} \\
& -\Big(s_{3,6}s_{4,5}(s_{1,2}-1)\mathcal{A}^{\phi^3}(B.22a) - s_{3,5}s_{4,6}(s_{1,2}-1)\mathcal{A}^{\phi^3}(B.22b) + (123456 \to 561234)\Big) \\
& -\Big(s_{1,6}s_{2,5}(s_{3,4}-1)\mathcal{A}^{\phi^3}(B.22c) - s_{1,5}s_{2,6}(s_{3,4}-1)\mathcal{A}^{\phi^3}(B.22d)\Big) \\
& - s_{1,4}s_{2,6}s_{3,5}\mathcal{A}^{\phi^3}(B.23a) - s_{1,5}s_{2,4}s_{3,6}\mathcal{A}^{\phi^3}(B.23b) - s_{1,3}s_{2,5}s_{4,6}\mathcal{A}^{\phi^3}(B.23c) \\
& + s_{1,5}s_{2,3}s_{4,6}\mathcal{A}^{\phi^3}(B.24a) + s_{1,3}s_{2,6}s_{4,5}\mathcal{A}^{\phi^3}(B.24b) + s_{1,6}s_{2,4}s_{3,5}\mathcal{A}^{\phi^3}(B.24c) \\
& - s_{1,6}s_{2,3}s_{4,5}\mathcal{A}^{\phi^3}(B.25) + s_{1,4}s_{2,5}s_{3,6}\mathcal{A}^{\phi^3}(B.26) \,.
\end{aligned}
\tag{17}
$$

In the second line, we include one cyclic image (with legs 7 and 8 fixed) of two terms, resulting in a total of four terms. Note that since we have two special legs 7 and 8, most of the terms cannot be related by cyclic rotation. The corresponding $\delta$ shift, as we have labeled for each term, can be found in Appendix B.2.

While the kinematical shift of $s_{i,j}$, *i.e.* the corresponding $d_{i,j}$ (minus the exponent of the PT factor) in (7), is directly given by the Pfaffian expansion, determining the combinatorial intersection (*i.e.* the kinematical shift in terms of the shift of $X_{a,b}$) can be a tedious task. Nevertheless, as we have shown in (B.16), the range of the kinematical shift $\delta_{a,b} = x_{a,b}$ is given by:

$$
x_{a,b} \in [-1,\, b-a-1]\,,
\tag{18}
$$

if we restrict $-2 \le d_{i,j} \le 0$. Note that for a triangulation that contains chord $(a, b)$, $u_{a,b}$ is proportional to $y_{a,b}$, so the minimum of $x_{a,b}$ implies that the residue at the pole $y_{a,b} = 0$ involves at most a first-order derivative. In particular, for the scaffolding chord $(a, b) = (2i-1, 2i+1)$, we naively expect $x_{2i-1,2i+1} \in [-1, 1]$. However, this estimate is too loose for the particular object we are considering. In fact, due to the overall factor $1/(z_{1,2}z_{3,4}\cdots z_{2n-1,2n})$, we have $x_{2i-1,2i+1} = 1 + d_{2i-1,2i}$ with $d_{2i-1,2i} = -1$ or $-2$, and thus

$$
x_{2i-1,2i+1} = 0,\ \text{or } -1\,.
\tag{19}
$$

Therefore the scaffolding residue that we will consider shortly for each term in the expansion (13) is given by:

$$
\operatorname*{Res}_{y_{2i-1,2i+1}=0} \mathcal{A}^{\phi^3}(\{\delta_{a,b}\}) =
\begin{cases}
\displaystyle \lim_{y_{2i-1,2i+1}\to 0} \mathcal{A}^{\phi^3}(\{\delta_{a,b}\}), & \delta_{2i-1,2i+1} = 0\,, \\[2ex]
\displaystyle \lim_{y_{2i-1,2i+1}\to 0}\left(\frac{\partial}{\partial y_{2i-1,2i+1}} y_{2i-1,2i+1}\mathcal{A}^{\phi^3}(\{\delta_{a,b}\})\right), & \delta_{2i-1,2i+1} = -1\,.
\end{cases}
\tag{20}
$$

## 2.2 A new formula for the $n$-gluon superstring amplitude from scaffolding residues

Upon scaffolding with residues $X_{1,3} = X_{3,5} = \ldots = X_{2n-1,1} = 0$, we obtain the $n$-gluon amplitude in scalar-scaffolded form in superstring theory (we switch the notation and denote the $n$-gon after scaffolding as $(1, 2, \ldots, n)$ instead of $(1, 3, 5, \ldots, 2n-1)$):

$$\mathcal{A}_n^{\text{super.}} = \sum_{\rho \subset [n-1]}^{N_n} T_\rho \; \mathcal{A}_n^{\text{mixed bos.}}(\rho_s | \bar{\rho}_g), \tag{21}$$

where the sum is taken over all products of cycles, denoted by $\rho$, chosen from $[n-1] := \{1, 2, \ldots, n-1\}$, *i.e.*, over all possible sub-cycles of the ordered set $\{1, 2, \ldots, n-1\}$ (the leg $n$ is fixed); all such products can be extracted abstractly from the expansion of the determinant of a $n \times n$ symmetric matrix, with total number $N_n = 2, 5, 17, 73, 388, \ldots$ for $n = 3, 4, 5, 6, 7, \ldots$ (OEIS A002135); *e.g.* for $n = 3$, $\rho = \emptyset, (12)$, for $n = 4$, $\rho = \emptyset, (12), (13), (23), (123)$, and similarly for $n = 5$, 17 such products including 13 single cycles (6 length-2 ones, 4 length-3 ones and 3 length-4 ones which are permutations of $T_{(1234)}$), and double-cycles such as $(12)(34), (13)(24), (14)(23)$. Here the mixed bosonic string amplitudes are defined in the Appendix A.1, and the integrand for multi-trace of scalar in $\rho$ are the product of Parke-Taylor factors, and each gluon in $\bar{\rho}$ forms the those $V$ and $W$'s building blocks.

We will give a worldsheet derivation of these formulas, but we emphasize that this differs from any formula we know of: it has only one special leg $n$ and it relates the superstring amplitude with other (mixed) bosonic string amplitudes. With the knowledge of the gauge invariance of mixed bosonic string amplitudes, it manifests the gauge invariance in this expansion of those gauge invariant building blocks $T_\rho$ as we will see below. It has much less terms than (13) since upon scaffolding residues, we identify two punctures $z_{2i-1}$ and $z_{2i}$ to $z_i$, makes the around $2^{2i}$ terms in the $2n$-gon cycle to be 1 term $[12\cdots i]$ in the $n$-gon. For example, $[123456]$ and other 7 terms by permutation $(12), (34), (56)$ become single term $[123]$. We comment that this new formula (21) is from a special choice of the $2n$-scalar superstring amplitude as we fix the two $(-1)$ ghost charge operators in the position $(2n-1, 2n)$. Upon scaffolding residues, this two $(-1)$ ghost charge operators should OPE to a new operator with the ghost charge $(-2)$, which makes the leg $n$ so special! If we change the choice to fix the two $(-1)$ ghost charge operators in the position such as $(2i, 2j)$, after scaffolding, it should obtain the familiar formula for $n$-point gluon superstring amplitude with the two special legs $(i, j)$.

For now, let us show that $T_\rho$ is nothing but a simple *nested commutator* of those $S_\alpha$ contributing to this term (thus a polynomial of Mandelstam variables/planar variables of the $2n$-gon on scaffolding residues): more precisely by shrinking the segment $(2i-1, 2i)$ to a vertex that we call $i$, any cycle factor (of $2n$-gon), $C_\alpha$ reduces to a product of cycles of $n$ gluons (where the 2-cycles become fixed points that we remove to define $\rho$):

$$T_\rho = \sum_{\alpha:\, C_\alpha \to \rho} \text{sgn}'(\alpha | \rho) S_\alpha \Big|_{s_{1,2}, s_{3,4}, \ldots, s_{2n-1,2n} \to 0}, \tag{22}$$

where the sum is over all $\alpha$ such that $C_\alpha$ reduces to $\rho$ (plus fixed points), and $\text{sgn}'(\alpha | \rho)$ are some signatures dependent on $\alpha$ and $\rho$. In Appendix A.2, we induce the explicit formula of $T_\rho$, which expands as the sum of the products of the chain of $s_{i,j}$'s commutators

$$T_\rho = \prod_{\text{single-cycle } \sigma \text{ in } \rho} T_\sigma, \qquad T_{(\sigma_1 \sigma_2 \cdots \sigma_m)} = s_{2\sigma_1-1],[2\sigma_2} s_{2\sigma_2-1],[2\sigma_3} \cdots s_{2\sigma_m-1],[2\sigma_1}, \tag{23}$$

for the special case of $m = 2$, we define $T_{(\sigma_1 \sigma_2)} = \frac{1}{2} s_{2\sigma_1-1],[2\sigma_2} s_{2\sigma_2-1],[2\sigma_2}$. Remarkably, upon translating to polarizations of the gluons, the polynomial $T_\rho$ becomes exactly a product of

traces of field-strengths for $\rho$! Note that $f_i^{\mu\nu} = k_i^\mu \epsilon_i^\nu - k_i^\nu \epsilon_i^\mu$ for $i$ in the gluon $n$-gon. We can identify the polarizations and momentum in the $2n$-gon kinematics by the relation $\epsilon_i = p_{2i-1}$, and $k_i = p_{2i-1} + p_{2i}$. For example,[3]

$$
\begin{aligned}
T_{(12)} &= s_{2,3}s_{1,4} - s_{1,3}s_{2,4} \simeq \epsilon_1 \cdot k_2 \epsilon_2 \cdot k_1 - \epsilon_1 \cdot \epsilon_2 k_1 \cdot k_2 = \mathrm{tr}(f_1 f_2)\,, \\
T_{(123)} &= s_{1,6}s_{2,4}s_{3,5} - s_{1,4}s_{2,6}s_{3,5} - s_{1,5}s_{2,4}s_{3,6} + s_{1,4}s_{2,5}s_{3,6} \\
&\quad - s_{1,6}s_{2,3}s_{4,5} + s_{1,3}s_{2,6}s_{4,5} + s_{1,5}s_{2,3}s_{4,6} - s_{1,3}s_{2,5}s_{4,6} \simeq \mathrm{tr}(f_1 f_2 f_3)\,.
\end{aligned}
\tag{24}
$$

A simple proof of (23) and $T_{(i_1 i_2 \cdots i_m)} \simeq \mathrm{tr}(f_{i_1} f_{i_2} \cdots f_{i_m})$ can be found in Appendix A.2.

Finally, the mixed bosonic string amplitudes contains scalars in the corresponding multicycle structure $\rho$, as well as the remaining gluons in the complement $\bar{\rho}$ (including $n$ which is always a gluon). For example, for $n = 3$,

$$
\mathcal{A}_3^{\mathrm{super.}} = \mathcal{A}_3^{\mathrm{bos.}}(\{1,2,3\}_g) + T_{(12)}\mathcal{A}_3((12)_s|\{3\}_g)\,,
\tag{25}
$$

and for $n = 4$, 5 terms including 4-gluon case $\mathcal{A}_4^{\mathrm{bos.}}$, 3 terms of the form $T_{(12)}\mathcal{A}_4((12)_s|\{3,4\}_g)$ (with 2 gluons) and a term with 1 gluon $T_{(123)}\mathcal{A}_4((123)_s|4_g)$.

$$
\begin{aligned}
\mathcal{A}_4^{\mathrm{super.}} &= \mathcal{A}_4^{\mathrm{bos.}}(\{1,2,3,4\}_g) + T_{(12)}\mathcal{A}_4((12)_s|\{3,4\}_g) + T_{(13)}\mathcal{A}_4((13)_s|\{2,4\}_g) \\
&\quad + T_{(23)}\mathcal{A}_4((23)_s|\{1,4\}_g) + T_{(123)}\mathcal{A}_4((123)_s|\{4\}_g)\,.
\end{aligned}
\tag{26}
$$

We will give the definition of such mixed bosonic string amplitudes using both curve integrals and worldsheet integrals. Independent of any representations, these mixed amplitudes can be obtained by acting *transmuted operators* [26] on the $n$-gluon bosonic string amplitude, $\mathcal{A}_n^{\mathrm{bos.}}$, similar to universal expansions for field-theory amplitudes [27].

We define two transmuted operators, for simplicity here we use the $n$-gluon kinematics, but one can always translate to $2n$-gon scalar kinematics.

$$
D_{i,j} = \partial_{\epsilon_i \cdot \epsilon_j}\,, \qquad D_{i,l,j} = \partial_{\epsilon_l \cdot k_i} - \partial_{\epsilon_l \cdot k_j} \equiv \partial_{\epsilon_l \cdot (k_i - k_j)}\,.
\tag{27}
$$

It is straightforward to show that acting the transmuted operators on the integrand of the pure bosonic string amplitude yields the integrand of the mixed bosonic string amplitude, *i.e.*, the product of Parke-Taylor factors and the remaining gluon correlators, since

$$
D_{i,j}W_{a,b} = \frac{\delta_{i,a}\delta_{j,b}}{z_{i,j}^2}\,, \qquad D_{i,l,j}V_a = \frac{\delta_{l,a}}{z_{l,i}} - \frac{\delta_{l,a}}{z_{l,j}} = \frac{\delta_{l,a}z_{i,j}}{z_{l,i}z_{l,j}}\,,
\tag{28}
$$

where $\delta_{i,j}$ is the Kronecker delta. See the explicit definitions of $V$ and $W$ in Appendix A.1. Here, we already observe that $D_{i,l,j}$ acting on $V_l$ gives rise to a soft factor $\frac{z_{i,j}}{z_{l,i}z_{l,j}}$, which serves as an intermediate step in forming a larger Parke-Taylor factor. For example:

$$
\begin{aligned}
[12] &= D_{1,2}W_{1,2}\,, & [123] &= D_{1,3}D_{1,2,3}(W_{1,3}V_2)\,, \\
[1234] &= D_{1,4}D_{1,2,4}D_{2,3,4}(W_{1,4}V_2 V_3)\,, & [12][34] &= D_{1,2}D_{3,4}(W_{1,2}W_{3,4})\,.
\end{aligned}
\tag{29}
$$

We now demonstrate how the transmuted operators can be used to derive mixed bosonic string amplitudes from the pure bosonic string amplitude. Let $\rho = \{\beta_1, \ldots, \beta_s\}$, the corresponding transmuted operator is defined as

$$
\hat{D}_\rho = \prod_{i=1}^{s} \left( D_{\beta_i[1],\beta_i[m_i]} \prod_{j=2}^{m_i-1} D_{\beta_i[j-1],\beta_i[j],\beta_i[m_i]} \right) \equiv \prod_{i=1}^{s} \hat{D}_{(\beta_i)}\,,
\tag{30}
$$

---

[3]Here, we have absorbed a factor $1/2$ in the definition of $\mathrm{tr}(f_1 f_2)$, other traces are defined as usual.

where $m_i$ denotes the length of $\beta_i$, and $\beta_i[j]$ denotes the $j$-th element of $\beta_i$.

In the case of a single trace $\alpha = (\alpha_1, \alpha_2, \ldots, \alpha_m)$, the associated transmuted operator takes the form

$$\hat{D}_{(\alpha)} = D_{\alpha_1,\alpha_m} \prod_{i=2}^{m-1} D_{\alpha_{i-1},\alpha_i,\alpha_m}. \tag{31}$$

Acting $\hat{D}_{(\alpha)}$ on the bosonic string correlator (A.3) yields

$$\hat{D}_{(\alpha)} \mathcal{I}_n^{\text{bos}} = \frac{1}{\alpha' z_{\alpha_1,\alpha_m}^2} \left( \prod_{i=2}^{m-1} \frac{z_{\alpha_m,\alpha_{i-1}}}{z_{\alpha_i,\alpha_m} z_{\alpha_i,\alpha_{i-1}}} \right) \mathcal{I}_{n|\alpha}^{\text{minor bos}} = -\text{PT}(\alpha) \mathcal{I}_{n|\alpha}^{\text{minor bos}}, \tag{32}$$

where $\mathcal{I}_{n|\alpha}^{\text{minor bos}}$ denotes the coefficient of the $V_{\alpha_i}$'s in $\mathcal{I}_n^{\text{bos}}$, i.e., $\mathcal{I}_{n|\alpha}^{\text{minor bos}} := \prod_{i \in \alpha} \partial_{V_i} \mathcal{I}_n^{\text{bos}}$. The action of the transmuted operator on $\mathcal{I}_{n|\alpha}^{\text{minor bos}}$ preserves the same structure as in (32). Applying this recursively, we obtain

$$\hat{D}_\rho \mathcal{I}_n^{\text{bos}} = \left( \prod_{\alpha \in \rho} \text{PT}(\alpha) \right) \mathcal{I}_{n|\rho}^{\text{minor bos}}, \tag{33}$$

where we omit the overall minus sign in (32) to maintain consistency in the definition. The RHS of the above equation exactly matches the definition of the mixed correlator given in (A.5). Thus, the $n$-gluon superstring amplitude can be expressed as

$$\mathcal{A}_n^{\text{super.}} = \left( \sum_{\rho \in [n-1]} T_\rho \hat{D}_\rho \right) \mathcal{A}_n^{\text{bos.}}. \tag{34}$$

Notably, specific dimensional reduction [28–30] of both sides of (21), or equivalently (34), gives rise to a new expansion of superstring-induced nonlinear sigma model (NLSM) amplitudes in terms of bosonic-string-induced NLSM+$\phi^3$ mixed amplitudes [30–32].

# 3   Properties and new structures of superstring amplitudes

Having derived a new formula for the $n$-gluon superstring amplitude, which is not only manifestly gauge invariant (given the gauge invariance of the bosonic-string building blocks) but also symmetric in $n-1$ legs, we now study interesting properties (old and new) of superstring amplitudes using it. Let us first comment on several very basic ones, which already illustrate certain non-trivial points of the new formula. First of all, one can check that, in terms of $2n$-gon kinematics, the combination of multi-linearity and gauge invariance of the scalar-scaffolded superstring amplitude, (13), still works in a similar way as the bosonic string case [20], though it requires some extra works since the prefactors $S_\alpha$ also plays a role in the derivation. On the other hand, (21) actually makes this property manifest term-by-term recursively: for any given term in (21), since this works for the scaffolding residue on any length-2 cycle (exactly as the bosonic-string case), we only need to check it for those legs in the prefactor $T_\rho$, which is manifestly gauge invariant (and linear in the polarization)!

Moreover, let us comment on another important property, which was derived for bosonic string amplitudes in [20], namely the factorization on any gluon pole with $X_{a,b} \to 0$. We immediately see a new complication: while one can repeat the derivation just as the bosonic string case for (13), it becomes rather complicated to recognize the resulting two sub-amplitudes as superstring amplitudes. Note that the worldsheet supersymmetry is reflected in the freedom of choosing a scalar pair such as $(2n-1, 2n)$ (or gluon $n$), which is exactly what makes the

proof of factorization more intricate: one has to restore this redundancy on one of the two sub-amplitude which requires the use of integration-by-parts identities! Similarly it is also nontrivial to prove gluon factorization for (21): on any pole $X_{a,b} \to 0$, the residue for some mixed amplitudes on the RHS correspond to a scalar factorization, and it is highly nontrivial that the linear combination factorizes by the exchanging a gluon when combined with traces of field-strengths, $T_\rho$. We leave a systematic study of such factorization properties to future works.

In the remainder of the section, we will briefly study interesting new features of superstring amplitudes from (13) and (21), which differ significantly from bosonic string amplitudes, such as the cancellation of $F^3$ vertices for leading singularities(LS), and the absence of tachyon poles. Furthermore, we will view (21) as new relations between the $n$-gluon superstring amplitude and its bosonic-string building blocks, to all orders in the $\alpha'$ expansion, and discuss the new relations for closed superstring amplitudes from double copy.

## 3.1 Cancellations of $F^3$ vertices, tachyon poles, *etc.*

As shown in [20], for any triangulation of the $n$-punctured surface (dual to a trivalent graph with $n$ external legs), the leading singularity of bosonic string amplitude can be computed as a $(n-3)$-fold residue of the curve integral (after taking scaffolding residues), which results in a combination of gluing of 3-pt YM and $F^3$ vertices. This gives a purely combinatorial/graphical way of computing any leading singularity to all orders in 't Hooft coupling for bosonic string amplitudes; in particular for any tree-level LS (one of the Catalan number of triangulations of $n$-gon, labeled by chords $(\alpha_1, \ldots, \alpha_{n-3})$), after taking $n$ scaffolding residues with $X_{1,3} = \ldots = X_{2n-1,1} = 0$, we obtain LS by taking $n-3$ residues of the form $X_{\alpha_1} = \ldots = X_{\alpha_{n-3}} = 0$, which in turn is given by the residue $y_{\alpha_1} = \ldots = y_{\alpha_{n-3}} = 0$ of the $n$-gluon bosonic string amplitude; in practice as explained in [20], this amounts to take the coefficient of $\prod y$ in the expansion of the Koba-Nielsen, $\prod u^X$, factor. The result is a nice combination of $2^{n-2}$ terms with different orders in $\alpha'$: since for each triangle of the triangulation (3-vertex of the graph) we have the 3-pt Yang-Mills (YM) amplitude plus the $F^3$ amplitude with an extra $\alpha'$; after gluing we have the pure YM LS, and the one with a single $F^3$ insertion at $\mathcal{O}(\alpha')$, and that with two $F^3$ insertions at $\mathcal{O}(\alpha'^2)$, ..., and the pure $F^3$ LS at $\mathcal{O}(\alpha'^{n-2})$. Remarkably, our formula guarantees that for any leading singularity, these higher-order terms with $F^3$ insertions must all cancel out, leaving only the pure YM term which is the correct LS for superstring amplitude!

Let us look at the simplest example for $n = 3$ where no residue needs to be taken (except for the scaffolding ones):

$$\mathcal{A}_3^{\text{super.}} = \mathcal{A}_3^{\text{bos.}}(\{1,2,3\}_g) + T_{(12)}\mathcal{A}_3((12)_s|3_g) = A_3^{\text{YM}} + \text{tr}(f_1 f_2 f_3) - 2\text{tr}(f_1 f_2)\epsilon_3 \cdot k_1 = A_3^{\text{YM}}. \quad (35)$$

Here the bosonic string amplitude contains two terms: the Yang-Mills 3-point amplitude $A_3^{\text{YM}}$ and the $F^3$ 3-point amplitude $\text{tr}(f_1 f_2 f_3)$. At 3-point kinematics, the mixed amplitude $\mathcal{A}_3((12)s|3_g) = -2\epsilon_3 \cdot k_1$, multiplied by the prefactor $T_{(12)} = \text{tr}(f_1 f_2) = \epsilon_1 \cdot k_2 \epsilon_2 \cdot k_1 - \epsilon_1 \cdot \epsilon_2 k_1 \cdot k_2$, yields $-\text{tr}(f_1 f_2 f_3)$. This precisely cancels the $F^3$ term in the bosonic string amplitude, as expected.

For $n = 4$, we compute the leading singularity at $s_{1,2} = 0$:

$$\begin{aligned}
\operatorname*{Res}_{s_{1,2}=0} \mathcal{A}_4^{\text{super.}} = {} & \operatorname*{Res}_{s_{1,2}=0} \mathcal{A}_4^{\text{bos.}}(\{1,2,3,4\}_g) + \operatorname*{Res}_{s_{1,2}=0} T_{(12)}\mathcal{A}_4((12)_s|\{3,4\}_g) \\
& + \operatorname*{Res}_{s_{1,2}=0} T_{(13)}\mathcal{A}_4((13)_s|\{2,4\}_g) + \operatorname*{Res}_{s_{1,2}=0} T_{(23)}\mathcal{A}_4((23)_s|\{1,4\}_g) \\
& + \operatorname*{Res}_{s_{1,2}=0} T_{(123)}\mathcal{A}_4((123)_s|\{4\}_g)
\end{aligned}$$

$$= \left( \mathrm{LS}_4^{YM|YM} + \mathrm{LS}_4^{YM|F^3} + \mathrm{LS}_4^{F^3|YM} + \mathrm{LS}_4^{F^3|F^3} \right) - \left( \mathrm{LS}_4^{F^3|YM} + \mathrm{LS}_4^{F^3|F^3} \right) - \mathrm{LS}_4^{YM|F^3}$$

$$= \mathrm{LS}_4^{YM|YM}. \tag{36}$$

Here, we use $\mathrm{LS}^{A|B}$ to denote the leading singularity at $s_{1,2} = 0$, with the left vertex given by $A$ and the right vertex by $B$. The leading singularity of the 4-point bosonic string amplitude consists of four terms. The residue of $T_{(12)}\mathcal{A}_4((12)_s|\{3,4\}_g)$ contributes to $\mathrm{LS}_4^{F^3|YM} + \mathrm{LS}_4^{F^3|F^3}$, since its left part is $2\mathrm{tr}(f_1 f_2)\epsilon_3 \cdot k_1$, which on the support of $s_{1,2} = 0$, becomes $\mathrm{tr}(f_1 f_2 f_3)$; and the right part remains the 3-point bosonic string amplitude, which includes contributions from both YM and $F^3$ vertices. The residues of the three terms on the second line yield the remaining $\mathrm{LS}_4^{YM|F^3}$. In total, all contributions from $F^3$ vertices cancel out, and analogous telescopic cancellations happen for higher $n$.

Similarly we should see that our formula guarantees the absence of tachyon poles for the superstring amplitudes. Note that every term on the RHS of (21) contains tachyon poles at $X_{a,b} = 1$, and it is intriguing to see that the residues at each such pole cancel out in the full combination. For example, the 4-point bosonic string amplitude has the explicit expression at finite $\alpha'$, $\mathcal{A}_4^{\mathrm{bos.}}(\{1,2,3,4\}_g) = B(1,2,3,4)F_{1234}^{(2)}$. The definitions of $B(1,2,3,4)$ and $F$-integral can be found in the literature [33, 34]. Now, we can use the transmuted operators (27) to obtain the mixed bosonic string amplitudes that appear in the expansion. For example:

$$\mathcal{A}_4((12)_s|\{3,4\}_g) = D_{1,2}B(1,2,3,4),$$

$$\mathcal{A}_4((123)_s|\{4\}_g) = D_{1,3}D_{1,2,3}B(1,2,3,4) = \frac{k_3 \cdot \epsilon_4}{s_{1,2}} + \frac{k_2 \cdot \epsilon_4 + k_3 \cdot \epsilon_4}{s_{2,3}}. \tag{37}$$

Note that $B(1,2,3,4) = A_4^{\mathrm{YM}} + \alpha'(\dots)$, and only the single-gluon mixed amplitudes are free of higher-order $\alpha'$ corrections. The function $B(1,2,3,4)$ also contains tachyon poles; however, after summing over all mixed bosonic string amplitudes, we recover the well-known 4-point superstring amplitude $\mathcal{A}_4^{\mathrm{super.}} = A_4^{\mathrm{YM}}F_{1234}^{(2)}$ [15, 35, 36], with all tachyon poles canceling out.

## 3.2 New relations for the $\alpha'$ expansions

We emphasize that (21) is a new relation between superstring amplitudes and various bosonic string mixed amplitudes, which hold for finite $\alpha'$, and it immediately implies infinite numbers of relations for $\alpha'$-expansion of these amplitudes. In the $\alpha' \to 0$ limit, each mixed bosonic-string amplitude reduces to mixed amplitudes with Yang-Mills coupled to (bi-adjoint) $\phi^3$ [34]: these amplitudes and their stringy corrections from bosonic strings have been studied in [37, 38]. Note that the leading order has mass dimension of $s^{2+n_{\mathrm{tr}}-n_s}$ where $n_{\mathrm{tr}}$ and $n_s$ are the number of traces and scalars respectively, and the trace factor carries mass dimension $s^{n_s}$ (in the $2n$-gon notation, the polarization carries the same mass dimension as the momentum), thus each term in RHS of (21) has mass dimension $2 + n_{\mathrm{tr}}$. This means that only the original bosonic-string term ($\alpha_0$) with $n_{\mathrm{tr}} = 0$ contributes to the leading order of superstring amplitude, or the $n$-gluon Yang-Mills amplitude (with mass dimension $s^2$), which is the same $\alpha' \to 0$ limit of both superstring and bosonic-string amplitudes. Now at the next, $\mathcal{O}(\alpha')$ order with mass dimension $s^3$, the LHS is 0 while bosonic-string amplitude reduces to YM amplitude with a single $F^3$ insertion, and we have

$$A_n^{\mathrm{YM}+F^3} = \sum_{\rho,\, n_{\mathrm{tr}}=1} T_\rho A_n^{\mathrm{YMS}}(\rho_s|\bar\rho_g). \tag{38}$$

For example, for $n = 4$ we have:

$$A_4^{\mathrm{YM}+F^3} = T_{(12)}\mathcal{A}_4((12)_s|\{3,4\}_g) + T_{(13)}\mathcal{A}_4((13)_s|\{2,4\}_g)$$

$$+ T_{(23)}\mathcal{A}_4((23)_s|\{1,4\}_g) + T_{(123)}\mathcal{A}_4((123)_s|\{4\}_g). \tag{39}$$

For $n = 5$ the result reads:

$$A_5^{\text{YM}+F^3} = T_{(12)}\mathcal{A}_5((12)_s|\{3,4,5\}_g) + 5 \text{ perms} + T_{(123)}\mathcal{A}_5((123)_s|\{4,5\}_g) + 3 \text{ perms} \tag{40}$$
$$+ T_{(1234)}\mathcal{A}_5((1234)_s|\{5\}_g) + T_{(1243)}\mathcal{A}_5((1243)_s|\{5\}_g) + T_{(1324)}\mathcal{A}_5((1324)_s|\{5\}_g),$$

where the 5 perms compress single trace terms $(13),(14),(23),(24),(34)$, and the 3 perms compress single trace terms $(134),(234),(124)$. Note that the double trace $T_{(12)(34)}$, $T_{(23)(14)}$, and $T_{(13)(24)}$ do not contribute. We have explicitly verified that our new formula (38) agrees with the result obtained from the CHY formula [39] up to 5 points.

### 3.3 Double copy: Heterotic and type II closed superstring

We have already discussed the new formula for the open superstring (21) from various aspects, so now let us briefly turn to the closed superstring via double copy.

The heterotic superstring amplitude arises from the double copy of the type I superstring and the bosonic string, while the type II superstring amplitude comes from the double copy of two type I superstrings. The derivation is similar to that for the type I superstring in Appendix A.2, but with the addition of the bosonic/super parts in the $\bar{z}$ sector.

The new formulas for the heterotic and type II closed superstring amplitudes are

$$\mathcal{M}_n^{\text{heterotic.}} = \sum_{\rho \subset [n-1]}^{N_n} T_\rho \, \mathcal{M}_n^{\text{mixed bos.}}(\rho_g|\bar{\rho}_h),$$
$$\mathcal{M}_n^{\text{super.}} = \sum_{\rho,\sigma \subset [n-1]}^{N_n} T_\rho \tilde{T}_\sigma \, \mathcal{M}_n^{\text{mixed bos.}}((\rho \cap \sigma)_s|(\bar{\rho} \cap \sigma)_g \cup (\rho \cap \bar{\sigma})_g|(\bar{\rho} \cap \bar{\sigma})_h), \tag{41}$$

where $\mathcal{M}$ denotes the closed string amplitudes, $h$ denotes the graviton, and $\tilde{T}$ denotes the trace over the polarization vectors $\tilde{\epsilon}$.

In the $\alpha' \to 0$ expansion, at the next-to-leading order we can obtain the double-copy version of (38), corresponding to gravity amplitudes with a single $R^2$ or $R^3$ insertion [39, 40]:

$$M_n^{\text{GR}+R^2} = \sum_{\rho,\, n_{\text{tr}}=1} T_\rho M_n^{\text{EYM}}(\rho_g|\bar{\rho}_h),$$
$$M_n^{\text{GR}+R^3} = \sum_{\rho,\sigma,\, n_{\text{tr}}=1}^{N_n} T_\rho \tilde{T}_\sigma \, M_n^{\text{EYMS}}((\rho \cap \sigma)_s|(\bar{\rho} \cap \sigma)_g \cup (\rho \cap \bar{\sigma})_g|(\bar{\rho} \cap \bar{\sigma})_h), \tag{42}$$

where the LHS denote the amplitudes with a single $R^2$ or $R^3$ insertion, corresponding to the next-to-leading order terms in the bosonic closed string amplitude. The building blocks on the RHS are Einstein-Yang-Mills-(Scalar) (EYM(S)) amplitudes, which are transmuted from gravity amplitudes [26]. One can further decompose these EYM(S) amplitudes into YMS amplitudes using the KLT relations [41]/color-kinematic duality and double copy [42–44], or compute them directly via the CHY formulae [28,45]. We have also explicitly verified those formula (42) agrees with the result obtained from the CHY formula [39] up to 4 points.

## 4  Conclusions and outlook

In this paper we have derived, first in the worldsheet formalism and then in terms of curve integrals, new formulas for tree-level superstring amplitudes. Inspired by the scaffolding picture, we first obtain (13) which express the superstring amplitude for $n$ pairs of scalars (or a special component of $2n$-gluon superstring amplitude) as a linear combination of $(2n-3)!!$

stringy $\text{Tr}(\phi^3)$ amplitudes with different kinematic shifts given by *combinatorial intersections* (which correspond to bosonic-string amplitudes with different cycle structures). Very nicely, upon scaffolding residues, this leads to a new formula for the $n$-gluon superstring amplitude, which is manifestly symmetric in $n-1$ gluons, (21): it is given by a sum of mixed bosonic string amplitudes with scalars and gluons, dressed by gauge-invariant *nested commutators*, (22), which become exactly traces of field-strengths. We have also discussed applications of our formulas, including the study of cancellation for tachyon poles and $F^3$ vertices, and more importantly new relations between superstring and bosonic string amplitudes to all orders in their $\alpha'$ expansions; in particular, at the first non-trivial order, (38) provides a new formula for gluon amplitude with a single $F^3$ insertion.

Our preliminary study opens up numerous avenues for future investigations, and let us mention a few of them. First of all, it would be much more satisfying to understand how the worldsheet supersymmetry could naturally emerge from "surfaceology" (without referring to worldsheet picture). For example, could we derive (21) directly from considerations in terms of curve integrals? Relatedly, we have not proved the cancellation of $F^3$ vertices in leading singularities to all $n$, which might provide more combinatorial understanding of our formulas. Note that many properties such as factorizations on gluon poles and cancellation of tachyon poles now depend on gauge fixing and integration-by-parts identities, and it would be interesting to prove them explicitly. Moreover, each term in our formula clearly has similar pattern of zeros and splits near zeros as the bosonic string, and it would be nice to work this and and compare with results of [13].

Another direction concerns external fermions and spacetime supersymmetry: we have only focused on external gluons, which naturally arises from scaffolding scalars. Is it possible to produce fermions in a similar way (see [46, 47] for related works)? Alternatively, along the line of [37, 38], one could further expand mixed bosonic string amplitudes into combinations of kinematic prefactors multiplied by amplitudes with more scalars and less gluons, in a way similar to the "universal expansion" for both tree- and one-loop amplitudes [27, 48]. It would be highly desirable to find a way of "uplifting" such prefactors to BRST invariants, perhaps based on some pure-spinor considerations [49–51] (for a review see [52]).

More concretely, our new formula (21) has provided new relations between superstring and bosonic string amplitudes to all orders in the $\alpha'$ expansion. The first non-trivial order gives a new formula for gluon amplitudes with a single $F^3$ insertion (see [39] for a CHY formula), but it would be very interesting to systematically spell out higher-order relations(see [53] for Berends-Giele currents in $\text{YM} + F^3 + F^4$ theory and [54] for effective Lagrangian of the open bosonic string). They involve various amplitudes with higher-dimensional operators as well as those in $DF^2 + \text{YM}$ theory [34]. Moreover, we have briefly commented on heterotic/type II amplitudes via double copy, and it would be highly desirable to study new relations for these closed-string amplitudes more systematically. Another aspect of the new relations concerns four-dimensional helicity (or supersymmetric) amplitudes from string theory: while the bosonic string amplitudes involve more complicated configurations such as all-plus and single-minus amplitudes, they must vanish when combined to superstring amplitudes. It would be interesting to study such four-dimensional string amplitudes both from the worldsheet and from the curve-integral formulation.

Last but not least, we have focused on tree-level amplitudes, but clearly it is worth considering generalizations to loops. Neither superstring or bosonic string amplitudes for higher-genus surfaces have been understood in the curve-integral formulation, and it would be extremely interesting already to translate building blocks of one-loop amplitudes in this language [48], or use the single-cut to reconstruct higher-loop integrand [3, 55]. At tree-level, the mixed YMS amplitudes have also important as the boundary data of the supergluon scattering in the AdS space [56, 57]. Even though we do not expect similar relations for loop-level string ampli-

tudes, our results again indicate the importance of mixed amplitudes also at loop level, which interpolate between stringy $\text{Tr}(\phi^3)$ amplitudes with $\prod \frac{dy}{y}$ and the gluon case with $\prod \frac{dy}{y^2}$. We plan to study curve integrals for such mixed loop amplitudes with external gluons and scalars in the future.

# Acknowledgments

We thank Nima Arkani-Hamed, Carolina Figueiredo and Jaroslav Trnka for inspiring discussions.

**Funding information** The work of Q.C. is supported by the National Natural Science Foundation of China under Grant No. 123B2075. The work of S.H. has been supported by the National Natural Science Foundation of China under Grant No. 12225510, 12447101, and by the New Cornerstone Science Foundation through the XPLORER PRIZE.

# A Worldsheet derivations

In this appendix, we will review the basics of tree-level open string amplitudes in the worldsheet language, and derive the new formula from superstring amplitudes.

## A.1 Review of tree-level open string amplitudes

The tree-level open string amplitude is the correlator function in the disk surface. After color decomposition, the color ordering open string amplitude is defined as,

$$\mathcal{A}_n(1, 2, \ldots, n) = \int_{D(12\cdots n)} d\mu_n \langle \mathcal{O}_1 \mathcal{O}_2 \cdots \mathcal{O}_n \rangle = \int_{D(12\cdots n)} d\mu_n \times \mathcal{I}_n \times \text{KN}, \qquad \text{(A.1)}$$

where the integration domain denotes $D(12\cdots n) = \{-\infty < z_1 < z_2 < \ldots < z_n < \infty\}$, the measure denotes $d\mu_n = \frac{\prod_{i=1}^{n} dz_i}{\text{vol SL}(2,\mathbb{R})}$ with the volume of the $\text{SL}(2,\mathbb{R})$ symmetry can be canceled by the three fixed punctures at $(0, 1, \infty)$.

All the vertex operator has the structure $\mathcal{O}_j = V(z_j)e^{ip_j \cdot X(z_j)}$ with the corresponding momentum $p$. The universal part from the Wick contraction of the $e^{ip_j \cdot X(z_j)}$'s is the Koba-Nielsen factor, $\text{KN} = \prod_{i<j} z_{i,j}^{2\alpha' p_i \cdot p_j}$ with $z_{i,j} = z_i - z_j$. The integrand $\mathcal{I}_n$ is from the Wick contraction of the $V(z_j)$'s.

We are interested in those massless vertex operator with $p^2 = 0$, and we list some as followed.

For bosonic string, the operators are $V_J^a(z) \sim J^a(z)$ and $V_\epsilon(z) \sim \epsilon \cdot i\partial X(z)$. $J^a(z)$ is the Kac-Moody current with the adjoint index $a$ of the $SU(N)$ gauge group. The polarization vector $\epsilon$ satisfy the condition $\epsilon_i \cdot p_i = 0$.

For superstring, here we only consider the NS sector, so we list the operator with the ghost charge $-1, 0$. $V_\epsilon^{(-1)}(z) \sim \epsilon \cdot \psi(z)e^{-\phi(z)}e^{ip \cdot X(z)}$, and $V_\epsilon^{(0)}(z) \sim [\epsilon \cdot i\partial X(z) + 2\alpha'(p \cdot \psi)\epsilon \cdot \psi(z)]e^{ip \cdot X(z)}$.

With those vertex operators, we now consider the corresponding integrands.

The single trace $n$-point scalar bosonic string integrand is Parke-Taylor factor, from the single trace $\text{Tr}(T^{a_1} \cdots T^{a_n})$ of $\langle V_J^{a_1}(z_1) V_J^{a_2}(z_2) \cdots V_J^{a_n}(z_n) \rangle$,

$$\mathcal{I}_n^{\text{scalar}} = \frac{1}{z_{1,2}z_{2,3}\cdots z_{n,1}} = \text{PT}(1, 2, \ldots, n). \qquad \text{(A.2)}$$

For $n$-point gluon bosonic string integrand from $\langle V_\epsilon(z_1) V_\epsilon(z_2) \cdots V_\epsilon(z_n) \rangle$ is

$$\mathcal{I}_n^{\text{bos}} = \sum_{r=0}^{\lfloor n/2 \rfloor + 1} \sum_{\{g,h\},\{l\}} \prod_s^r W_{g_s,h_s} \prod_t^{n-2r} V_{l_t}, \tag{A.3}$$

where we have a summation over all partitions of $\{1, 2, \ldots, n\}$ into $r$ pairs $\{g_s, h_s\}$ and $n - 2r$ singlets $l_t$, each summand given by the product of $W$'s and $V$'s.

$$V_i := \sum_{j \neq i}^n \frac{\epsilon_i \cdot p_j}{z_{i,j}}, \qquad W_{i,j} := \frac{\epsilon_i \cdot \epsilon_j}{\alpha' z_{i,j}^2}. \tag{A.4}$$

For bosonic string, we can consider the mixed version of scalar and gluon vertex operators. The $n$-point $m$-trace($\{\alpha_1, \alpha_2, \ldots, \alpha_m\}$, each $\alpha$ is non-empty list) scalar and gluon integrand is from the multi trace $\text{Tr}(\alpha_1) \cdots \text{Tr}(\alpha_m)$ of the mixed correlator.

$$\mathcal{I}_n^{\text{mix.bos}}(\alpha_1, \alpha_2, \ldots, \alpha_m) = \prod_{i=1}^m \text{PT}(\alpha_i) \sum_{r=0}^{\lfloor (n-|\alpha|)/2 \rfloor + 1} \sum_{\{g,h\},\{l\}} \prod_s^r W_{g_s,h_s} \prod_t^{n-|\alpha|-2r} V_{l_t}, \tag{A.5}$$

where the second sum run over all partitions of $\{1, 2, \ldots, n\} / \bigcup_{i=1}^m \alpha_i$ into $r$ pairs $\{g_s, h_s\}$ and $n - 2r$ singlets $l_t$, and $|\alpha|$ denotes the total length of $\alpha_1, \alpha_2, \ldots, \alpha_m$.

For tree-level superstring, the total ghost charge must be $-2$, so we need two $V_\epsilon^{(-1)}(z)$'s and $n - 2$ $V_\epsilon^0(z)$'s to construct the $n$-point gluon superstring integrand. The worldsheet supersymmetry guarantees the integrand is independent with the position of two $V_\epsilon^{(-1)}(z)$'s. The gauge-fixed superstring correlator is defined (see eq. (4.8) of [58]) as

$$\mathcal{I}_n^{\text{super}} = \frac{1}{z_{i_0,j_0}} \sum_{q=0}^{\lfloor n/2 \rfloor - 1} \sum_{\substack{\text{distinct} \\ \text{pairs} \\ \{i_l, j_l\}}} \prod_{l=1}^q \left( \frac{-2\epsilon_{i_l} \cdot \epsilon_{j_l}}{\alpha' z_{i_l,j_l}^2} \right) \text{Pf} \Psi_{\{1,\ldots,n\} \setminus \{i_1, j_1, \ldots, i_q, j_q\}}^{[i_0,j_0]}, \tag{A.6}$$

where the second sum goes over all $q$ distinct unordered pairs $\{i_1, j_2\}, \{i_2, j_2\}, \ldots, \{i_q, j_q\}$ of labels from the set $\{1, 2, \ldots, n\} \setminus \{i_0, j_0\}$, and $(i_0, j_0)$ is arbitrary. Here we use $\Psi_{\{1,\ldots,n\} \setminus \{i_1, j_1, \ldots, i_q, j_q\}}^{[i_0,j_0]}$ to denote the $2(n-2q-1) \times 2(n-2q-1)$ matrix yielded by removing the $i_0$-th, $j_0$-th and the $i, j, n+i, n+j$-th columns and rows from $\Psi$ for each $(ij) \in \{i_1, j_1, \ldots, i_q, j_q\}$. The $\Psi_n$ matrix is an anti-symmetric $2n \times 2n$ matrix, defined as

$$\Psi_n = \begin{pmatrix} \mathbf{A}_n & -\mathbf{C}_n^{\text{T}} \\ \mathbf{C}_n & \mathbf{B}_n \end{pmatrix}, \tag{A.7}$$

where $\mathbf{A}_n, \mathbf{B}_n, \mathbf{C}_n$ are $n \times n$ matrices involving the polarization vectors, whose components are given as

$$(\mathbf{A}_n)_{ab} = \begin{cases} \frac{s_{a,b}}{z_a - z_b}, & a \neq b, \\ 0, & a = b, \end{cases} \quad (\mathbf{B}_n)_{ab} = \begin{cases} \frac{2\epsilon_a \cdot \epsilon_b}{z_a - z_b}, & a \neq b, \\ 0, & a = b, \end{cases} \quad (\mathbf{C}_n)_{ab} = \begin{cases} \frac{2\epsilon_a \cdot p_b}{z_a - z_b}, & a \neq b, \\ \sum_{c \neq a} \frac{2\epsilon_a \cdot p_c}{z_a - z_c}, & a = b, \end{cases} \tag{A.8}$$

and the reduced Pfaffian of $\Psi$ is defined as

$$\text{Pf}' \Psi_n := \frac{1}{z_{a,b}} \text{Pf}\left( \Psi_n^{[a,b]} \right), \tag{A.9}$$

where $a, b$ $(a, b \leq n)$ is arbitrary and $\Psi_n^{[a,b]}$ denotes the minor with the $a^{\text{th}}$ and $b^{\text{th}}$ rows and columns removed. As shown in [13], in order to simplify the superstring correlator (A.6), we define an differential operator $\widehat{O}_{a,b}$:

$$\widehat{O}_{a,b} := -\frac{1}{\alpha'}\epsilon_a \cdot \epsilon_b \partial_{\epsilon_a \cdot \epsilon_b} \partial_{s_{a,b}}. \tag{A.10}$$

Then we can induce the following compact formula for superstring integrand

$$\mathcal{I}_n^{\text{super}} = \prod_{\{a,b\}\subset\{1,\ldots,n\}\backslash\{i_0,j_0\}} (1 + \widehat{O}_{a,b})\, \text{Pf}'\Psi_n. \tag{A.11}$$

## A.2 Derivation of the new formula for superstring

The $2n$-scalar kinematical configuration [20] is

$$p_i \cdot \epsilon_j = 0, \qquad \forall (i,j) \in (1,\ldots,2n),$$
$$\epsilon_i \cdot \epsilon_j = \begin{cases} 1, & \text{if } (i,j) \in \{(1,2);(3,4);(5,6);\cdots;(2n-1,2n)\}, \\ 0, & \text{otherwise.} \end{cases} \tag{A.12}$$

Applying the $2n$-scalar kinematical configuration to (A.11) and choosing $(i_0, j_0) = (2n-1, 2n)$, we have

$$\mathcal{I}_{2n}^{\text{super}} \xrightarrow{(A.12)} \frac{1}{z_{2n-1,2n}} \prod_{i=1}^{n-1}(1 + \widehat{O}_{2i-1,2i})\text{Pf}'\mathbf{A}_{2n} \prod_{i=1}^{n-1}\frac{\epsilon_{2i-1}\cdot\epsilon_{2i}}{z_{2i-1,2i}}$$
$$= \prod_{i=1}^{n}\frac{1}{z_{2i-1,2i}} \prod_{i=1}^{n-1}(1 - \frac{1}{\alpha'}\partial_{s_{2i-1,2i}})\text{Pf}'\mathbf{A}_{2n} \tag{A.13}$$
$$= \prod_{i=1}^{n}\frac{1}{z_{2i-1,2i}} \left(\frac{1}{z_{2n-1,2n}}\text{Pf}\tilde{\mathbf{A}}_{2n}^{[2n-1,2n]}\right),$$

where we omit the factors 2 before each $\epsilon \cdot \epsilon$ and the $\tilde{\mathbf{A}}$ is defined in (10).[4]

In the following, we continue to scaffold and translate the momentum in the $2n$-gon kinematics to polarizations or momentum in the gluon $n$-gon. First, let us focus on the Pfaffian which can be expanded as

$$\text{Pf}\tilde{\mathbf{A}}_{2n}^{[2n-1,2n]} = \sum_{\substack{1=i_1<i_2<\cdots<i_{n-1} \\ i_1<j_1,\ldots,i_{n-1}<j_{n-1}}} \text{sgn}(i_1 j_1 i_2 j_2 \cdots i_{n-1}j_{n-1})\frac{\tilde{s}_{i_1,j_1}}{z_{i_1,j_1}}\frac{\tilde{s}_{i_2,j_2}}{z_{i_2,j_2}}\cdots\frac{\tilde{s}_{i_{n-1},j_{n-1}}}{z_{i_{n-1},j_{n-1}}}. \tag{A.14}$$

If we fix each $z_{\text{odd},\text{odd}+1}$ and identify the remaining $z_{\text{odd}+1}$ with $z_{\text{odd}}$, and set all $s_{\text{odd},\text{odd}+1}$ to zero, then those $z_{\text{odd}}$ have weight 2. Consequently, each term would be formed as Parke-Taylor factors multiplied by some Mandelstam variables, for example

$$\text{sgn}(\{o\})\frac{s_{o_1,o_2+1}s_{o_2,o_3+1}\cdots s_{o_m,o_1+1}}{z_{o_1,o_2+1}z_{o_2,o_3+1}\cdots z_{o_m,o_1+1}}\left(\prod_{i=m+1}^{n-1}\frac{\tilde{s}_{o_i,o_i+1}}{z_{o_i,o_i+1}}\right) \tag{A.15}$$

$$\rightarrow \text{sgn}(\{o\})s_{o_1,o_2+1}s_{o_2,o_3+1}\cdots s_{o_m,o_1+1}\text{PT}(o_1 o_2 \cdots o_m)\left(\prod_{i=m+1}^{n-1}\frac{-1}{\alpha' z_{o_i,o_i+1}}\bigg|_{z_{o_i+1}\to z_{o_i}}\right),$$

---

[4]In this Appendix, all factors of 2 in Lorentz products such as $2\epsilon_i \cdot \epsilon_j, 2\epsilon_i \cdot p_j, 2p_i \cdot p_j$ and the overall signature $(-1)^{n-1}$ are omitted, we also define $s_{i,j} = p_i \cdot p_j$ for convenience.

where $o_1, o_2, \ldots, o_{n-1} = 1, 3, \ldots, 2n-3$, and $\text{sgn}(\{o\})$ denotes the signature of the permutation $(o_1, o_2+1, o_2, \ldots, o_m, o_1+1, o_{m+1}, o_{m+1}+1, \ldots, o_{n-1}, o_n+1)$. It's evident that exchanging certain pairs of $o_i$ and $o_i + 1$ should yield the same Parke-Taylor factor, up to a sign determined by the permutation's signature. By collecting terms with same Parke-Taylor factor, we obtain

$$\text{sgn}(\{o\}) t_{(o_1 o_2 \cdots o_m)} \text{PT}(o_1 o_2 \cdots o_m) \left( \prod_{i=m+1}^{n-1} \frac{-1}{z_{o_i, o_i+1}} \bigg|_{\alpha' z_{o_i+1} \to z_{o_i}} \right), \tag{A.16}$$

$$t_{(o_1 o_2 \cdots o_m)} := s_{o_1], [o_2+1} s_{o_2], [o_3+1} \cdots s_{o_m], [o_1+1}. \tag{A.17}$$

The $t_{(o_1 o_2 \cdots o_m)}$ and $\text{PT}(o_1 o_2 \cdots e_m)$ share the same cyclic and reflection symmetries, and only $(m-1)!/2$ of permutations are independent. Remarkably, the nest commutator $t_{(o_1 o_2 \cdots o_m)}$ can be translated to traces of field strength for the gluons

$$s_{o_1], [o_2+1} s_{o_2], [o_3+1} \cdots s_{o_m], [o_1+1} = \text{tr}(p^{[\mu_1}_{o_1+1} p_{o_1, \mu_2]} p^{[\mu_2}_{o_2+1} p_{o_2, \mu_3]} \cdots p^{[\mu_m}_{o_m+1} p_{o_m, \mu_{m+1}]}), \tag{A.18}$$

where $p^{[\mu_1}_{o_1+1} p_{o_1, \mu_2]} := p^{\mu_1}_{o_1+1} p_{o_1, \mu_2} - p^{\mu_2}_{o_1+1} p_{o_1, \mu_1}$, and we using $k_i$ to denote the momentum of gluons. To identify the polarizations and momentum in the $2n$-gon kinematics by the relation $\epsilon_i = p_{2i-1}$, and $k_i = p_{2i-1} + p_{2i}$,

$$p^{[\mu_1}_{o_1+1} p^{\mu_2]}_{o_1} \simeq (p_{o_1} + p_{o_1+1})^{[\mu_1} p^{\mu_2]}_{o_1} \simeq k^{[\mu_1}_{(o_1+1)/2} \epsilon^{\mu_2]}_{(o_1+1)/2} = f^{\mu_1 \mu_2}_{(o_1+1)/2}. \tag{A.19}$$

The signature $\text{sgn}(\{o\})$ in (A.16) is only dependent on $m$

$$\begin{aligned}
&\text{sgn}(o_1, o_2 + 1, o_2, \ldots, o_m, o_1 + 1, o_{m+1}, o_{m+1} + 1, \ldots, o_{n-1}, o_{n-1} + 1) \\
&= -\text{sgn}(o_1 + 1, o_1, o_2 + 1, o_2, \ldots, o_{m+1}, o_{m+1} + 1, \ldots, o_{n-1}, o_{n-1} + 1) \\
&= (-1)^{m+1} \text{sgn}(o_1, o_1 + 1, o_2, o_2 + 1, \ldots, o_{n-1}, o_{n-1} + 1) \\
&= (-1)^{m+1},
\end{aligned} \tag{A.20}$$

where in the first equality, $o_1 + 1$ is moved to the first position, and in the second equality, the $m$ pairs of $(o_i + 1, o_i)$ are commuted. We have only talked about the single cycle before, and the multi-cycle is totally the same. Recovering the prefactors of Pfaffian in (A.13) and Koba-Nielsen factor we have

$$\left( \sum_{\rho} (-1)^{|\rho|} t_\rho \, \text{PT}_\rho \right) \text{KN} \prod_{o_i \in \bar{\rho}} \frac{-1}{\alpha' z^2_{o_i, o_i+1}} \prod_{i \in \rho} \frac{1}{z_{o_i, o_i+1}}, \tag{A.21}$$

where $\rho$ is the collection of the products of cycles from $(1, 3, \ldots, 2n-3)$, and $\bar{\rho}$ are the complementary of $\rho$. The last step is to solve the $n$ of scaffolding residues. In general, we have

$$\text{Res}_{z_{2i}=z_{2i-1}} \frac{\text{KN}}{z_{2i-1, 2i}} = \text{KN} \big|_{z_{2i} \to z_{2i-1}}, \tag{A.22}$$

and

$$\text{Res}_{z_{2i}=z_{2i-1}} \frac{\text{KN}}{\alpha' z^2_{2i-1, 2i}} = \frac{1}{\alpha'} \left( \partial_{z_{2i}} \text{KN} \right) \big|_{z_{2i} \to z_{2i-1}} = \left( \tilde{V}_{2i-1} \, \text{KN} \right) \big|_{z_{2i} \to z_{2i-1}}, \tag{A.23}$$

where

$$\tilde{V}_{2i-1} := \sum_{j \neq 2i-1, 2i, 2n} \frac{s_{2i, j}}{z_{2i, j}} \bigg|_{z_{2i} \to z_{2i-1}} = \sum_{j \neq i}^{n} \frac{p_{2i-1} \cdot (p_{2j-1} + p_{2j})}{z_{2i-1, 2j-1}} \simeq \sum_{j \neq i}^{n} \frac{\epsilon_i \cdot k_j}{z_{i, j}}. \tag{A.24}$$

Similarly, we define

$$\tilde{W}_{2i-1, 2j-1} = \frac{s_{2i-1, 2j-1}}{\alpha' z^2_{2i-1, 2j-1}} \simeq \frac{\epsilon_i \cdot \epsilon_j}{\alpha' z^2_{i, j}}. \tag{A.25}$$

Using the derivative relation $\partial_{z_{2j}} \tilde{V}_{2i-1}\big|_{z_{2j} \to z_{2j-1}} = \alpha' W_{i,j}$, for the single trace objects, we have

$$
\operatorname*{Res}_{z_{o_1+1}=z_{o_1}} \left( \cdots \left( \operatorname*{Res}_{z_{o_m+1}=z_{o_m}} \left( KN \prod_{i=1}^{m} \frac{-1}{\alpha' z_{o_i,o_i+1}^2} \right) \right) \cdots \right)
$$

$$
= \left( \frac{-1}{\alpha'} \right)^m \left( \prod_{i=1}^{m} \partial_{z_{o_i+1}} KN \right) \Bigg|_{z_{o_i+1} \to z_{o_i}} \tag{A.26}
$$

$$
= (-1)^m \left( \sum_{r=0}^{\lfloor m/2 \rfloor+1} \sum_{\{g,h\},\{l\}} \prod_{s}^{r} \tilde{W}_{g_s,h_s} \prod_{t}^{m-2r} \tilde{V}_{l_t} \right) KN \Bigg|_{z_{o_i+1} \to z_{o_i}} .
$$

The last line is exactly the mixed operator defined in (A.5). The Koba-Nielsen factor for $2n$-scalars is also identified to the Koba-Nielsen factor for $n$-gluons

$$
\prod_{i<j<2n} z_{i,j}^{\alpha' s_{i,j}} \Bigg|_{\substack{z_{2i} \to z_{2i-1} \\ s_{2i-1,2i} \to 0}} = \prod_{i=1}^{n-1} z_{2i-1,2n-1}^{\alpha' p_{2n-1} \cdot (p_{2i}+p_{2i-1})} \prod_{i<j<n} z_{2i-1,2j-1}^{\alpha'(p_{2i}+p_{2i-1}) \cdot (p_{2j}+p_{2j-1})}
$$

$$
= z_{2n-1}^{\alpha' p_{2n-1} \cdot \sum_{i=1}^{2n-2} p_i} \prod_{i<j<n} z_{2i-1,2j-1}^{\alpha'(p_{2i}+p_{2i-1}) \cdot (p_{2j}+p_{2j-1})}
$$

$$
= \prod_{i<j<n} z_{2i-1,2j-1}^{\alpha'(p_{2i}+p_{2i-1}) \cdot (p_{2j}+p_{2j-1})} \tag{A.27}
$$

$$
\simeq \prod_{i<j<n} z_{2i-1,2j-1}^{\alpha' k_i \cdot k_j} ,
$$

where we use the gauge fixing $z_{2n} \to \infty$ in the second line. Finally, we get the result after taking all scaffolding residues

$$
\mathcal{I}_n^{\text{super}} \simeq \sum_{\rho} T_\rho \, \mathcal{I}_n^{\text{mix.bos}}(\rho). \tag{A.28}
$$

## A.3 The closure of the new formula for superstring

Under the $2n$-scalar kinematical configuration (A.12), the new formula for superstring must regenerate the Pfaffian.

It's evident that only $W_{2i-1,2i}$ would contribute to the result while $V_i$ do not. Regarding the traces, since $\epsilon_{\text{odd}} \cdot \epsilon_{\text{odd}+1}$ occurs iff $f_{\text{odd}}$ and $f_{\text{odd}+1}$ are adjacent, only traces with even length would survive. Hence, we have

$$
\begin{cases}
V_i \to 0, & W_{o_i,o_i+1} \to \dfrac{1}{\alpha' z_{o_i,o_i+1}^2}, \\
(f_{o_i} \cdot f_{o_i+1})^{\mu_{2i-1}\mu_{2i}} \to -k_{o_i}^\mu k_{o_i+1}^\nu.
\end{cases} \tag{A.29}
$$

Considering a single trace $\operatorname{Tr}(f_{o_1} f_{o_1+1} \cdots f_{o_m} f_{o_m+1})$ and combining it with the Parke-Taylor factor we have

$$
PT(o_1, o_1+1, \ldots, o_m, o_m+1) \operatorname{Tr}(f_{o_1} f_{o_1+1} \cdots f_{o_m} f_{o_m+1}) \tag{A.30}
$$

$$
= PT(o_1, o_1+1, \ldots, o_m, o_m+1)(f_{o_1} \cdot f_{o_1+1})^{\mu_1\mu_2} \cdots (f_{o_m} \cdot f_{o_m+1})^{\mu_{2m-1}\mu_{2m}} \eta_{\mu_2\mu_3} \cdots \eta_{\mu_{2m}\mu_1}
$$

$$
\to (-1)^m PT(o_1, o_1+1, \ldots, o_m, o_m+1) k_{o_1}^{\mu_1} k_{o_1+1}^{\mu_2} \cdots k_{o_m}^{\mu_{2m-1}} k_{o_m+1}^{\mu_{2m}} \eta_{\mu_2\mu_3} \cdots \eta_{\mu_{2m}\mu_1}
$$

$$
= (-1)^m \frac{k_{o_1}^{\mu_1} k_{o_1+1}^{\mu_2}}{z_{o_1,o_1+1}} \cdots \frac{k_{o_m}^{\mu_{2m-1}} k_{o_m+1}^{\mu_{2m}}}{z_{o_m,o_m+1}} \frac{\eta_{\mu_2\mu_3}}{z_{o_1+1,o_2}} \cdots \frac{\eta_{\mu_{2m}\mu_1}}{z_{o_m+1,o_1}}
$$

$$
= \operatorname{sgn}(o_1+1, o_2, \ldots, o_m+1, o_1) \frac{1}{z_{o_1,o_1+1}} \cdots \frac{1}{z_{o_m,o_m+1}} \frac{k_{o_1+1} \cdot k_{o_2}}{z_{o_1+1,o_2}} \cdots \frac{k_{o_m+1} \cdot k_{o_1}}{z_{o_m+1,o_1}}.
$$

Other surviving orderings of $\mathrm{Tr}(f_{o_1} f_{o_1+1} \cdots f_{o_m} f_{o_m+1})$ can be obtained by $(m-1)!$ permutations for the positions of the $m$ pairs $f_{o_i} f_{o_i+1}$, or by $2^{m-1}$ permutations for every pair $(o_i, o_i + 1)$ individually.[5] The latter operation introduces a minus sign, while the former does not. Hence, the sign change under such permutations $\sigma$ is captured by $\mathrm{sgn}(\sigma)$.

By permuting the index of metrics $\eta$ in the first line of (A.30), one can also generate multi-trace structures. For example, $(f_{o_1} f_{o_1+1})^{\mu_1 \mu_2} (f_{o_2} \cdot f_{o_2+1})^{\mu_3 \mu_4} \eta_{\mu_1 \mu_2} \eta_{\mu_3 \mu_4} = \mathrm{Tr}(f_{o_1} f_{o_2}) \mathrm{Tr}(f_{o_3} f_{o_4})$. Therefore, summing over all the possible permutations is equivalent to summing over all independent multi-traces for $m$ pairs of adjacent field strength $f_{o_1} f_{o_1+1}, \dots, f_{o_m} f_{o_m+1}$. This summation can be organized as follows

$$\left( \sum_\sigma \mathrm{sgn}(\sigma) \prod_{i=1}^m \frac{k_{\sigma(o_i+1)} \cdot k_{\sigma(o_{i+1})}}{z_{\sigma(o_i+1), \sigma(o_{i+1})}} \right) \prod_{i=1}^m \frac{1}{z_{o_i, o_i+1}} \prod_{i=m+1}^n \frac{1}{\alpha' z_{o_i, o_i+1}^2} , \tag{A.31}$$

where we identify $z_{o_{m+1}}$ and $z_{o_1}$ in the first product, and $\sigma$ sum over all the possible unequal permutations. The $\alpha'$ dependent terms are from $W_{i,j}$. Extracting out the common factor $(1/z_{2n-1,2n}^2) \prod_{i=1}^{n-1} 1/z_{2i-1,2i}$, the summation is

$$\left( \sum_\sigma \mathrm{sgn}(\sigma) \prod_{i=1}^m \frac{k_{\sigma(o_i+1)} \cdot k_{\sigma(o_{i+1})}}{z_{\sigma(o_i+1), \sigma(o_{i+1})}} \right) \prod_{i=m+1}^{n-1} \frac{-\alpha'^{-1}}{z_{o_i+1, o_i}} . \tag{A.32}$$

Remarkably, deleting a pair of $(f_o f_{o+1})^{\mu\nu}$ from (A.30) is equivalent to extracting the coefficient of $k_o \cdot k_{o+1}$ in (A.32) and multiplying by $-\alpha'^{-1}$. Hence, the factor $-\alpha'^{-1}$ can be recursively absorbed by shifting $k_o \cdot k_{o+1} \to k_o \cdot k_{o+1} - \alpha'^{-1}$. Finally, we obtain

$$\sum_{\substack{1 = i_1 < i_2 < \cdots < i_{n-1} \\ i_1 < j_1, \dots, i_{n-1} < j_{n-1}}} \mathrm{sgn}(i_1 j_1 i_2 j_2 \cdots i_{n-1} j_{n-1}) \frac{\tilde{s}_{i_1, j_1}}{z_{i_1, j_1}} \frac{\tilde{s}_{i_2, j_2}}{z_{i_2, j_2}} \cdots \frac{\tilde{s}_{i_{n-1}, j_{n-1}}}{z_{i_{n-1}, j_{n-1}}} , \tag{A.33}$$

which is exactly the expansion of Pfaffian (A.14).

# B  Curve integrals: Review and derivations

## B.1  A lightening review of curve integrals for the disk

Recently, a new reformulation of the scattering amplitudes of $\mathrm{Tr}(\phi^3)$ and its extensions to other theories, such as Yang-Mills and the nonlinear sigma model was introduced in a series of papers [9, 10, 20]. This reformulation is known as the curve integral formalism which decries the scattering amplitudes as an integral over functions associated with curves on a given surface. This description is valid at all order for the 't Hooft topological expansion for $\mathrm{Tr}(\phi^3)$, and at least the leading planar limit at any loop for its extension.

For our purposes, we will focus on tree level, where the corresponding surface is a disk with $2n$ marked points on the boundary. To further specify the setup, we choose a scaffolding triangulation of the surface [9, 20] for later convenience, *i.e.*, a triangulation with chords $(1,3),(3,5),\dots,(2n-1,1)$ forming an inner $n$-gon (See the left panel of Figure 3 for an example with $2n = 6$). The triangulation is dual to a fat graph, which defines the path for the each curve $\mathcal{C}$ and encode the word and corresponding $u_\mathcal{C}$ variables.

---

[5]The cyclicity and reflection symmetries reduced the number of permutations form $m!$ and $2^m$ to $(m-1)!$ and $2^{m-1}$, respectively.

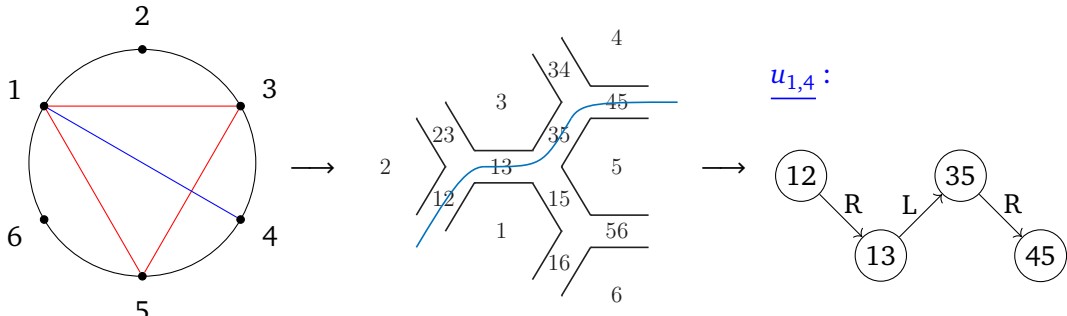

Figure 3: Scaffolding triangulation of a disk with 6 marked points (left), the corresponding fat graph (middle), and the word associated with the chord $(1,3)$ (right).

To be specific, at tree level, each curve is labeled by two endpoints $a, b$ on the boundary and is associated with the kinematical data $X_{a,b} = (p_a + p_{a+1} + \ldots + p_{b-1})^2$. To record the path, we begin by labeling the roads of the fat graph according to the sides they touch. Using this labeling, we can trace the path of a given curve along the fat graph by keeping track of the roads it traverses and whether it turns left or right at each intersection(See the middle of Figure 3).

We adopt the convention that each left turn corresponds to moving upward, while each right turn corresponds to moving downward. By following this rule, we construct a "mountainscape" uniquely associated with the curve – an object we refer to as the word (See the right panel of Figure 3).

In order to determine the $u$ variables associated with the word, we introduce two matrices:

$$M_L(y_{a,b}) = \begin{bmatrix} y_{a,b} & y_{a,b} \\ 0 & 1 \end{bmatrix}, \qquad M_R(y_{a,b}) = \begin{bmatrix} y_{a,b} & 0 \\ 1 & 1 \end{bmatrix}, \tag{B.1}$$

where $L$ and $R$ stand for left and right turns, respectively. We define $y_{a,a+1} := 1$ for boundary variables. We then multiply these matrices according to the word, i.e., each time the curve turns left or right at some road $y_{a,b}$. At the end of the process, we obtain a $2 \times 2$ matrix $M_{\mathcal{C}}$ with entries $m_{i,j}$ denoting the element in the $i$th row and $j$th column. The corresponding $u$ variable is then given by:

$$u_{\mathcal{C}} = \frac{m_{1,2} \cdot m_{2,1}}{m_{1,1} \cdot m_{2,2}}. \tag{B.2}$$

For instance, $M_{1,4}$ and $u_{1,4}$ in our 6-point example are given by:

$$M_R(1) M_L(y_{1,3}) M_R(y_{3,5}) = \begin{bmatrix} 1 & 0 \\ 1 & 1 \end{bmatrix} \begin{bmatrix} y_{1,3} & y_{1,3} \\ 0 & 1 \end{bmatrix} \begin{bmatrix} y_{3,4} & 0 \\ 1 & 1 \end{bmatrix} = \begin{bmatrix} y_{3,5} y_{1,3} + y_{1,3} & y_{1,3} \\ 1 + y_{3,5} y_{1,3} + y_{1,3} & 1 + y_{1,3} \end{bmatrix},$$

$$u_{1,4} = \frac{1 + y_{1,3} y_{3,5} y_{1,3}}{(1 + y_{1,3})(1 + y_{3,5})}. \tag{B.3}$$

One can proceed to compute all $2n(2n-3)/2$ $u$-variables associated with curves on the surface. The stringy $\mathrm{Tr}(\phi^3)$ amplitude is then given by

$$\mathcal{A}_{2n}^{\phi^3} = \int_{\mathbb{R}^{2n-3} > 0} \prod_J \frac{dy_J}{y_J} \prod_{(a,b)} u_{a,b}^{X_{a,b}}, \tag{B.4}$$

where the product over $J$ runs over the triangulation of the surface. For example:

$$
\begin{aligned}
\mathcal{A}_6^{\phi^3} = \int_{\mathbb{R}^3>0} \frac{dy_{1,3}dy_{3,5}dy_{1,5}}{y_{1,3}y_{3,5}y_{1,5}} & \left(\frac{y_{1,3}(1+y_{3,5})}{1+y_{1,3}+y_{1,3}y_{3,5}}\right)^{X_{1,3}} \left(\frac{1+y_{1,3}+y_{1,3}y_{3,5}}{(1+y_{1,3})(1+y_{3,5})}\right)^{X_{1,4}} \\
& \left(\frac{(1+y_{1,3})y_{1,5}}{1+y_{1,5}+y_{1,3}y_{1,5}}\right)^{X_{1,5}} \left(\frac{1+y_{1,3}}{1+y_{1,3}+y_{1,3}y_{3,5}}\right)^{X_{2,4}} \left(\frac{1+y_{1,5}+y_{1,3}y_{1,5}}{(1+y_{1,3})(1+y_{1,5})}\right)^{X_{2,5}} \\
& \left(\frac{1+y_{1,5}}{1+y_{1,5}+y_{1,3}y_{1,5}}\right)^{X_{2,6}} \left(\frac{(1+y_{1,5})y_{3,5}}{1+y_{3,5}+y_{1,5}y_{3,5}}\right)^{X_{3,5}} \left(\frac{1+y_{3,5}+y_{1,5}y_{3,5}}{(1+y_{1,5})(1+y_{3,5})}\right)^{X_{3,6}} \\
& \left(\frac{1+y_{3,5}}{1+y_{3,5}+y_{1,5}y_{3,5}}\right)^{X_{4,6}} .
\end{aligned}
\tag{B.5}
$$

Very nicely, it was shown in [9, 20] that a simple deformation of the stringy $\mathrm{Tr}(\phi^3)$ theory computes the YMS amplitudes with $n$ pairs of scalars in the bosonic string:

$$
\mathcal{A}_{2n}^{\mathrm{YMS}} = \int_{\mathbb{R}^{2n-3}>0} \prod_J \frac{dy_J}{y_J^2} \prod_{(a,b)} u_{a,b}^{X_{a,b}} .
\tag{B.6}
$$

Moreover, its scaffolding residues at $X_{1,3}, X_{3,5}, \ldots, X_{2n-1,1} = 0$, or equivalently the residues at $y_{1,3}, y_{3,5}, \ldots, y_{2n-1,1} = 0$ at the integrand level, effectively fuse the $n$ pairs of scalars into gluons, resulting in a pure gluon amplitude. The kinematical data in the effective $n$-gluon problem is given by $\epsilon_i = (1-\alpha)p_{2i} - \alpha p_{2i-1}$, and $k_i = p_{2i-1} + p_{2i}$ with $\alpha$ to be a free parameter related to the gauge redundancy. Gauge invariance and multilinearity are reflected in the following identities written in terms of planar variables:

$$
\begin{aligned}
\mathcal{A}_n^{\mathrm{gluon}} &= \sum_{j\neq\{2i-1,2i,2i+1\}} (X_{2i,j} - X_{2i-1,j}) \times \mathcal{Q}_{2i,j} \\
&= \sum_{j\neq\{2i-1,2i,2i+1\}} (X_{2i,j} - X_{2i+1,j}) \times \mathcal{Q}_{2i,j},
\end{aligned}
\tag{B.7}
$$

where $\mathcal{Q}_{2i,j} := \frac{\partial}{\partial X_{2i,j}} \mathcal{A}_n$ is independent of $X_{2i,j}$.

## B.2 Combinatorial intersection as kinematical shifts

We consider any monomial of $z_{i,j} := z_i - z_j$ for $i < j$ which has weight $-2$ for every $z_i$ variable, and by factoring out the Parke-Taylor factor, we have a *ratio* that can be written as a monomial of $n(n-3)/2$ $u_{a,b}$ variables:

$$
\prod_{i<j} z_{i,j}^{d_{i,j}} = \mathrm{PT}(1,2,\ldots,n) \prod_{(a,b)} u_{a,b}^{x_{a,b}},
\tag{B.8}
$$

where on the LHS, degrees $d_{i,j} = d_{j,i}$ satisfy $\sum_{j\neq i} d_{i,j} = -2$ for any $i$, and on the RHS, $\mathrm{PT}(1,\ldots,n) = \frac{1}{z_{1,2}\cdots z_{n-1,n}z_{n,1}}$, and $u_{a,b} = \frac{z_{a-1,b}z_{a,b-1}}{z_{a,b}z_{a-1,b-1}}$ (with degree denoted as $x_{a,b}$); note that while each $z_i$ is associated with the $i$-th edge of the $n$-gon, each chord $(a,b)$ (for non-adjacent $a < b$) connects vertices $a, b$ (we define vertex $a$ to be intersection of edge $a-1$ and $a$). In the following derivation, we assume that the differences $z_{i,j} = z_i - z_j$ always appear with $i < j$. However, since the Parke-Taylor factor includes a term $z_{n,1}$, there is an overall minus sign, which we will omit. We now proceed to derive the relations between the relations between $d_{i,j}$ and $x_{a,b}$.

By noticing that $x_{a,b}$ is the analog of $X_{a,b}$ and $d_{i,j}$ is the analog of $s_{i,j}$ except that for $s_{i,i+1}$ we have $d_{i,i+1}+1$ due to the PT factor, it is straightforward to see

$$
\begin{aligned}
d_{i,i+1}+1 &= x_{i,i+2}, && \text{for } i = 1,\ldots,n, \\
-d_{i,j} &= x_{i,j} + x_{i+1,j+1} - x_{i,j+1} - x_{i+1,j}, && \text{non-adjacent } i,j,
\end{aligned}
\tag{B.9}
$$

where $x_{i,i+1} = 0$; thus start from any monomial of $u$ variables (together with PT factor), we can go from RHS to LHS; by inverting these relations, we can also go from LHS to RHS and find that $x_{a,b}$ is given by the sum of $d_{i,j}$ for all pairs of edges $i,j$ between vertices $a, b$, or in the interval $\{a, a+1, \ldots, b-1\}$, plus all the $+1$ shift for $d_{a,a+1}, \ldots, d_{b-2,b-1}$:

$$
x_{a,b} + 1 = b - a + \sum_{a \le i < j < b} d_{i,j}.
\tag{B.10}
$$

To summarize, we have a combinatorial interpretation of the degree $x_{a,b}$ in the ratio (for non-adjacent $a < b$): it is given by $b-a-1$ plus the total degree of $z_{i,j}$ factors with all pairs $i,j$ in the interval $\{a, a+1, \ldots, b-1\}$. For the purpose of this paper, let us now assume $-2 \le d_{i,j} \le 0$, and it is straightforward to determine the range of $x_{a,b}$: Let $Q_{a,b} = \sum_{a \le i < j < b} d_{i,j}$ and $K = \{a, a+1, \ldots, b-1\}$ with cardinality $m = b - a$. Then the sum over all off-diagonal entries within $I$ satisfies:

$$
\sum_{i \in K} \sum_{\substack{j \in K \\ j \ne i}} d_{i,j} = \sum_{i \in K} \left( \sum_{j \ne i} d_{i,j} - \sum_{j \notin K} d_{i,j} \right) = -2m - \sum_{i \in K} \sum_{j \notin K} d_{i,j},
\tag{B.11}
$$

where in the second equality we have used $\sum_{j \ne i} d_{i,j} = -2$. Since the left-hand side equals $2Q_{a,b}$ we find:

$$
Q_{a,b} = -m - \frac{1}{2} \sum_{i \in K} \sum_{j \notin K} d_{i,j}.
\tag{B.12}
$$

Each term $d_{i,j}$ with $i \in K$, $j \notin K$, appears only once in this sum. As the total row sum of $d_{i,j}$ is fixed at -2, it follows that

$$
\sum_{j \notin K} d_{i,j} \in [-2, 0], \quad \text{for each } i \in K,
\tag{B.13}
$$

and thus the total interaction term satisfies

$$
\sum_{i \in K} \sum_{j \notin K} d_{i,j} \in [-2m, 0].
\tag{B.14}
$$

We conclude that the possible values of $Q_{a,b}$ lie within the interval

$$
Q_{a,b} \in [-m, 0], \quad \text{where} \quad m = b - a.
\tag{B.15}
$$

And therefore

$$
x_{a,b} \in [-1, b - a - 1].
\tag{B.16}
$$

Let us present some explicit examples: for $n = 4$, the corresponding ratios for $z_{1,2}^{-1} z_{3,4}^{-1} z_{1,3}^{-1} z_{2,4}^{-1}$, $z_{1,3}^{-1} z_{1,4}^{-1} z_{2,3}^{-1} z_{2,4}^{-1}$, $z_{1,2}^{-2} z_{3,4}^{-2}$, $z_{1,3}^{-2} z_{2,4}^{-2}$ are given by: $u_{2,4}, u_{1,3}, u_{2,4}/u_{1,3}, u_{1,3} u_{2,4}$ respectively. For monomial $z_{1,2}^{-2} z_{3,4}^{-1} z_{3,5}^{-1} z_{4,5}^{-1}$ at 5-point, we have the following counting (see Table 1) for ratios in terms of $u$'s:

Table 1: Combinatorial intersection for $z_{1,2}^{-2}z_{3,4}^{-1}z_{3,5}^{-1}z_{4,5}^{-1}$.

| $(a,b)$ | $b-a-1$ | $\sum_{a\leq i<j<b} d_{i,j}$ | $x_{a,b}$ |
|---------|---------|------------------------------|-----------|
| (1,3)   | 1       | -2                           | -1        |
| (1,4)   | 2       | -2                           | 0         |
| (2,4)   | 1       | 0                            | 1         |
| (2,5)   | 2       | -1                           | 1         |
| (3,5)   | 1       | -1                           | 0         |

Therefore we have:

$$\frac{z_{1,2}^{-2}z_{3,4}^{-1}z_{3,5}^{-1}z_{4,5}^{-1}}{\text{PT}(1,2,3,4,5)} = \frac{u_{2,4}u_{2,5}}{u_{1,3}}. \tag{B.17}$$

And similarly for monomial $z_{1,3}z_{3,5}z_{2,5}z_{2,4}z_{1,4}$ the combinatorial intersection reads (see Table 2):

Table 2: Combinatorial intersection for $z_{1,3}z_{3,5}z_{2,5}z_{2,4}z_{1,4}$.

| $(a,b)$ | $b-a-1$ | $\sum_{a\leq i<j<b} d_{i,j}$ | $x_{a,b}$ |
|---------|---------|------------------------------|-----------|
| (1,3)   | 1       | -1                           | 1         |
| (1,4)   | 2       | -1                           | 1         |
| (2,4)   | 1       | 0                            | 1         |
| (2,5)   | 2       | -1                           | 1         |
| (3,5)   | 1       | 0                            | 1         |

As a consequence, we have:

$$\frac{z_{1,3}z_{3,5}z_{2,5}z_{2,4}z_{1,4}}{\text{PT}(1,2,3,4,5)} = u_{1,3}u_{1,4}u_{2,4}u_{2,5}u_{3,5}. \tag{B.18}$$

For $n = 6$, there are nine $u$ variables relevant to the counting problem. For example, the combinatorial intersection of the term $z_{1,2}^{-1}z_{3,4}^{-1}z_{5,6}^{-2}z_{1,3}^{-1}z_{2,4}^{-1}$ is given by (see Table 3):

$$\frac{z_{1,2}^{-1}z_{3,4}^{-1}z_{5,6}^{-2}z_{1,3}^{-1}z_{2,4}^{-1}}{\text{PT}(1,2,3,4,5,6)} = \frac{u_{2,4}u_{2,6}u_{3,6}u_{4,6}}{u_{1,5}}. \tag{B.19}$$

Applying this algorithm, we also find the following results:

$$\frac{z_{1,2}^{-1}z_{3,4}^{-1}z_{5,6}^{-2}z_{1,4}^{-1}z_{2,3}^{-1}}{\text{PT}(1,2,3,4,5,6)} = \frac{u_{2,6}u_{3,6}u_{4,6}}{u_{1,5}}, \tag{B.20}$$

$$\frac{z_{1,3}^{-1}z_{3,5}^{-1}z_{2,5}^{-1}z_{2,6}^{-1}z_{4,6}^{-1}z_{1,4}^{-1}}{\text{PT}(1,2,3,4,5,6)} = u_{1,3}u_{1,4}u_{1,5}u_{2,4}u_{2,5}^{2}u_{2,6}u_{3,5}u_{3,6}u_{4,6}. \tag{B.21}$$

It is straightforward to extend the analysis to higher points. Below, we present some additional non-trivial examples for $n = 8$, which give rise to the $\delta$-shift in (17):

Table 3: Combinatorial intersection for $z_{1,2}^{-1} z_{3,4}^{-1} z_{5,6}^{-2} z_{1,3}^{-1} z_{2,4}^{-1}$.

| $(a, b)$ | $b - a - 1$ | $\sum_{a \leq i < j < b} d_{i,j}$ | $x_{a,b}$ |
|---|---|---|---|
| (1,3) | 1 | -1 | 0 |
| (1,4) | 2 | -2 | 0 |
| (1,5) | 3 | -2 | -1 |
| (2,4) | 1 | -1 | 1 |
| (2,5) | 2 | -2 | 0 |
| (2,6) | 3 | -2 | 1 |
| (3,5) | 1 | -1 | 0 |
| (3,6) | 2 | -1 | 1 |
| (4,6) | 1 | 0 | 1 |

$$\frac{z_{1,2}^{-2} z_{3,4}^{-1} z_{5,6}^{-1} z_{7,8}^{-2} z_{3,6}^{-1} z_{4,5}^{-1}}{\text{PT}(1,2,\ldots,8)} = \frac{u_{2,4} u_{2,5} u_{2,6} u_{2,8} u_{4,8} u_{5,8} u_{6,8}}{u_{1,3} u_{1,7} u_{3,7}}, \tag{B.22a}$$

$$\frac{z_{1,2}^{-2} z_{3,4}^{-1} z_{5,6}^{-1} z_{7,8}^{-2} z_{3,5}^{-1} z_{4,6}^{-1}}{\text{PT}(1,2,\ldots,8)} = \frac{u_{2,4} u_{2,5} u_{2,6} u_{2,8} u_{4,6} u_{4,8} u_{5,8} u_{6,8}}{u_{1,3} u_{1,7} u_{3,7}}, \tag{B.22b}$$

$$\frac{z_{1,2}^{-1} z_{3,4}^{-2} z_{5,6}^{-1} z_{7,8}^{-2} z_{1,6}^{-1} z_{2,5}^{-1}}{\text{PT}(1,2,\ldots,8)} = \frac{u_{1,4} u_{2,4} u_{2,8} u_{3,8} u_{4,6} u_{4,7} u_{4,8}^2 u_{5,8} u_{6,8}}{u_{1,7} u_{3,5}}, \tag{B.22c}$$

$$\frac{z_{1,2}^{-1} z_{3,4}^{-2} z_{5,6}^{-1} z_{7,8}^{-2} z_{1,5}^{-1} z_{2,6}^{-1}}{\text{PT}(1,2,\ldots,8)} = \frac{u_{1,4} u_{2,4} u_{2,6} u_{2,8} u_{3,8} u_{4,6} u_{4,7} u_{4,8}^2 u_{5,8} u_{6,8}}{u_{1,7} u_{3,5}}, \tag{B.22d}$$

$$\frac{z_{1,2} z_{3,4}^{-1} z_{5,6}^{-1} z_{7,8}^{-2} z_{1,4}^{-1} z_{2,6}^{-1} z_{3,5}^{-1}}{\text{PT}(1,2,\ldots,8)} = \frac{u_{1,4} u_{2,4} u_{2,5} u_{2,6} u_{2,8} u_{3,8} u_{4,6} u_{4,7} u_{4,8}^2 u_{5,8} u_{6,8}}{u_{1,7}}, \tag{B.23a}$$

$$\frac{z_{1,2} z_{3,4}^{-1} z_{5,6}^{-1} z_{7,8}^{-2} z_{1,5}^{-1} z_{2,4}^{-1} z_{3,6}^{-1}}{\text{PT}(1,2,\ldots,8)} = \frac{u_{1,4} u_{2,4} u_{2,6} u_{2,8} u_{3,6} u_{3,8} u_{4,6} u_{4,7} u_{4,8}^2 u_{5,8} u_{6,8}}{u_{1,7}}, \tag{B.23b}$$

$$\frac{z_{1,2} z_{3,4}^{-1} z_{5,6}^{-1} z_{7,8}^{-2} z_{1,3}^{-1} z_{2,5}^{-1} z_{4,6}^{-1}}{\text{PT}(1,2,\ldots,8)} = \frac{u_{2,4} u_{2,5} u_{2,6} u_{2,8} u_{3,6} u_{3,8} u_{4,6} u_{4,8} u_{5,8} u_{6,8}}{u_{1,7}}, \tag{B.23c}$$

$$\frac{z_{1,2} z_{3,4}^{-1} z_{5,6}^{-1} z_{7,8}^{-2} z_{1,5}^{-1} z_{2,3}^{-1} z_{4,6}^{-1}}{\text{PT}(1,2,\ldots,8)} = \frac{u_{2,6} u_{2,8} u_{3,6} u_{3,8} u_{4,6} u_{4,8} u_{5,8} u_{6,8}}{u_{1,7}}, \tag{B.24a}$$

$$\frac{z_{1,2} z_{3,4}^{-1} z_{5,6}^{-1} z_{7,8}^{-2} z_{1,3}^{-1} z_{2,6}^{-1} z_{4,5}^{-1}}{\text{PT}(1,2,\ldots,8)} = \frac{u_{2,4} u_{2,5} u_{2,6} u_{2,8} u_{3,8} u_{4,8} u_{5,8} u_{6,8}}{u_{1,7}}, \tag{B.24b}$$

$$\frac{z_{1,2} z_{3,4}^{-1} z_{5,6}^{-1} z_{7,8}^{-2} z_{1,6}^{-1} z_{2,4}^{-1} z_{3,5}^{-1}}{\text{PT}(1,2,\ldots,8)} = \frac{u_{1,4} u_{2,4} u_{2,8} u_{3,8} u_{4,6} u_{4,7} u_{4,8}^2 u_{5,8} u_{6,8}}{u_{1,7}}, \tag{B.24c}$$

$$\frac{z_{1,2} z_{3,4}^{-1} z_{5,6}^{-1} z_{7,8}^{-2} z_{2,3}^{-1} z_{4,5}^{-1} z_{1,6}^{-1}}{\text{PT}(1,2,\ldots,8)} = \frac{u_{2,8} u_{3,8} u_{4,8} u_{5,8} u_{6,8}}{u_{1,7}}, \tag{B.25}$$

$$\frac{z_{1,2} z_{3,4}^{-1} z_{5,6}^{-1} z_{7,8}^{-2} z_{1,4}^{-1} z_{2,5}^{-1} z_{3,6}^{-1}}{\text{PT}(1,2,\ldots,8)} = \frac{u_{1,4} u_{2,4} u_{2,5} u_{2,6} u_{2,8} u_{3,6} u_{3,8} u_{4,6} u_{4,7} u_{4,8}^2 u_{5,8} u_{6,8}}{u_{1,7}}. \tag{B.26}$$

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
