# Peer review of "Superstring amplitudes meet surfaceology"

_SciPost Physics, doi:SciPost Phys. 19, 114 (2025)_

## Round 1 · Referee Report · Anonymous (Referee 1) · 2025-8-26

Report

This paper derives a new formula for the tree-level gluon scattering amplitudes in type-I superstring theory. The final expression is presented in terms of mixed gluon–scalar amplitudes in bosonic string theory. The formula has several nice features, such as manifest symmetry and gauge invariance. It is obtained by exploiting recent advances in the “curve-integral” formulation of Tr\phi^3 amplitudes and their intriguing connections with gluon amplitudes.

The authors further study various non-trivial properties of the new formula, including the cancellation of the tachyon pole in bosonic string amplitudes and the vanishing of contributions from higher-derivative terms F^3; these are some basic properties of superstring amplitudes. Finally, they extend the discussion to closed superstring amplitudes by applying the double-copy construction.

Overall, I find the paper novel, interesting, and suitable for publication in SciPost. That said, there are some small typos and points where the presentation could be improved for clarity. My comments/questions are as follows:

  1. On page 1, first sentence: “... to all loops and to all orders in the ’t Hooft coupling...” is somewhat confusing, since loop expansion usually coincides with the expansion in the ’t Hooft coupling. Does “all loops” here instead refer to the genus expansion in color?

  2. Page 4: introducing the abbreviation “Yang-Mills (YM)” may not be necessary since it was introduced on page 1 already. Similar comments apply to " leading singularity (LS)" on page 13, the word "leading singularity" appears a few times before introducing "LS".

  3. Three lines after equation (2.2): the sentence containing “... with the usual.” seems incomplete and should be revised.

  4. After equation (2.2): the notation [I] is defined as a Parke-Taylor factor, but since this is used frequently later, it would be helpful to present it explicitly as an equation. Related to this, a different notation for Parke-Taylor factors was used in equation (A.2). It may be better to use a single notation.  

  5. After equation (2.15): the phrase “A_8^{bos.} with a factor (s−1)(s−1)(s−1)...” might be better phrased as “with a factor of the form (s−1)(s−1)(s−1).” The same applies to the following sentence.

  6. After equation (2.20): “... selected from [n−1] (so leg n is fixed).”: It is not clear whether [n−1] has been defined. I assume it does not denote a Parke-Taylor factor, which I mentioned earlier. 

  7. Equation (2.32): the period at the end should be a comma. Similar punctuation issues occur elsewhere and should be checked.

  8. Finally, it is known that in the \alpha'-expansion of superstring amplitudes, each order contains numbers of uniform transcendentality, whereas bosonic string amplitudes include numbers of lower transcendentality. Since the new formula expresses superstring amplitudes in terms of bosonic amplitudes, can this uniform transcendentality property be seen here? Presumably this is related to the cancellation of F^3 discussed in the paper?

Recommendation

Publish (easily meets expectations and criteria for this Journal; among top 50%)

  • validity: top
  • significance: high
  • originality: top
  • clarity: good
  • formatting: excellent
  • grammar: good

Author:  Qu Cao  on 2025-09-30  [id 5874]

(in reply to Report 1 on 2025-08-26)
Category:
answer to question

Thank you for the comments. Let me answer these comments. 1. For real Riemann surfaces, the all-loop expansion does not coincide with the expansion in the ’t Hooft coupling. We recommend the paper [arXiv:2309.15913] for details. 2. We have updated these abbreviations in the revised version. 3. This sentence has been revised accordingly. 4. The PT factor has been revised and is now presented explicitly in the form of an equation. 5. These sentences have been phrased accordingly. 6. The definition of $[n-1]$ has been revised. 7. The comma in the equation has been corrected. 8. As you mentioned, the bosonic string amplitudes include terms of lower transcendentality. In the literature, we have $A^{\text{super}} = A^{\text{YM}} \times F, \quad A^{\text{bos}} = B \times F$, where B is a function of $\alpha'$. In our new relation eq.(2.33), we find $A^{\text{YM}} = \sum_\rho D_\rho T_\rho B$. This means that all combinations of these B functions reproduce the Yang-Mills field-theory amplitudes, which indicates the uniform transcendentality. This result arises from the cancellation of the F^3 terms and tachyon poles.

---

## Round 1 · Referee Report · Anonymous (Referee 2) · 2025-9-23

Report

In this paper, the authors present a new formula for superstring amplitudes with $n$-gluons. The formula is based on the ``surface'' description of scattering amplitudes that was recently introduced. Via the surface description, bosonic string amplitudes were constructed prior to this paper via a shift of Tr $\phi^3$ amplitudes. This paper extends this to a new formulation of $n$-gluon superstring amplitudes and their new formula has some very nice features:

  1. It is manifestly gauge-invariant.
  2. It is manifestly permutation invariant in $n{-}1$ labels.
  3. It is a sum over bosonic string amplitudes, giving a new relation between super and bosonic string amplitudes.

The paper is well-written and provides a lot of examples that make it easy to understand notation in formulae.

In addition to the points raised in the other report, I was wondering if the authors could add a few comments on the following:

  1. Under equation (2.33) the authors mention that there is a dimensional reduction that would give amplitudes of a ``superstring-induced'' NLSM. Since the supersymmetrizations of NLSMs are quite tightly constrained, could the authors explain what their proposed model is?

  2. The relation between these double-ordered bosonic amplitudes and single ordered superstring amplitudes is interesting. Can these new relations between bosonic and superstring amplitudes be seen as a result of monodromy relations or as coming from a KLT or KLT-like relation?

  3. The discussion in the first paragraph on page 10 about ghosts is a bit confusingly worded. From the text, it seems that if one fixes the last two legs to be the two (-1) ghost charge operators, then leg $n$ is clearly special. Could the authors clarify why leg $n$ is special if one is interested in the second case i.e. $(2i,2j)$ which on the scaffolding residue gives $n$ gluons?

These minor comments aside, I believe this paper is a great addition to the surfaceology literature and recommend it for publication on SciPost.

Recommendation

Publish (surpasses expectations and criteria for this Journal; among top 10%)

  • validity: top
  • significance: high
  • originality: good
  • clarity: high
  • formatting: perfect
  • grammar: -

Author:  Qu Cao  on 2025-09-30  [id 5873]

(in reply to Report 2 on 2025-09-23)
Category:
answer to question

Thank you for the comments. Let me answer these questions. 1. The NLSM model we suggest here is obtained from the superstring after a special dimensional reduction (e.g., imposing $\epsilon_i \cdot \epsilon_j = p_i \cdot p_j$ and $\epsilon_i \cdot k_j = 0$). We refer to this as the superstring-induced NLSM model, which, however, is well-defined only in the NS (bosonic) sector. For more details, we recommend [arXiv:2404.11648]. 2. As far as we know, we have no clue how this new relation could be derived from monodromy or KLT relations. 3. From the structure of vertex operators in different ghost pictures, if one chooses two adjacent vertex operators both in the (-1) picture, then through the OPE (or equivalently by taking the scaffolding residue), these two operators combine into a new vertex operator in the (-2) picture. In the second case, say for (2i,2j), the vertex operator at position 2i in the (-1) picture together with the vertex operator at position 2i-1 in the (0) picture combine into a vertex operator at position i in the (-1) picture. The same holds for (2j). Thus, in this case, the resulting integrand still involves two vertex operators in the (-1) picture, which is precisely the standard integrand discussed in the GSW textbook. In contrast, our new formula involves a single vertex operator in the (-2) picture, chosen at position n. We recommend the paper [arXiv:2506.15299] for the explict expression of the vertex operator in the (-2) picture.

---

## Round 2 · List of Changes

1. We have updated these abbreviations for YM and LS, in the revised version. 2. Three lines after equation (2.2): This sentence has been revised accordingly. 3. After equation (2.2): The PT factor has been revised and is now presented explicitly in the form of an equation. 4. After equation (2.15): These sentences have been phrased accordingly. 5. After equation (2.20): The definition of [n-1] has been revised. 6. Equation (2.32): The comma in the equation has been corrected.

---

## Editorial Decision

published